# XDH-1 inactivation causes xanthine stone formation in *Caenorhabditis elegans* which is inhibited by SULP-4-mediated anion exchange in the excretory cell

Jennifer Snoozy[1], Sushila Bhattacharya[1], Brandon Johnson[1], Robin R. Fettig[1,2], Ashley Van Asma[1], Chloe Brede[1], Sophia G. Miller[3], Martina Ralle[3], Kurt Warnhoff [1,4]*

**1** Pediatrics and Rare Diseases Group, Sanford Research, Sioux Falls, South Dakota, United States of America, **2** Department of Basic Biomedical Sciences, Sanford School of Medicine, University of South Dakota, Vermillion, South Dakota, United States of America, **3** Department of Molecular and Medical Genetics, Oregon Health & Science University, Portland, Oregon, United States of America, **4** Department of Pediatrics, Sanford School of Medicine, University of South Dakota, Sioux Falls, South Dakota, United States of America

* kurt.warnhoff@sanfordhealth.org

## Abstract

Xanthine dehydrogenase (XDH) is a molybdenum cofactor (Moco) requiring enzyme that catabolizes hypoxanthine into xanthine and xanthine into uric acid, the final steps in purine catabolism. Human patients with mutations in XDH develop xanthinuria which can lead to xanthine stones in the kidney, recurrent urinary tract infections, and renal failure. Currently, there are no therapies for treating human XDH deficiency. Thus, understanding mechanisms that maintain purine homeostasis is an important goal of human health. Here, we used the nematode *Caenorhabditis elegans* to model human XDH deficiency using two clinically relevant paradigms: Moco deficiency or loss-of-function mutations in *xdh-1*, the *C. elegans* ortholog of XDH. Both Moco deficiency and *xdh-1* loss of function caused the formation of autofluorescent xanthine stones in *C. elegans*. Surprisingly, only 2% of *xdh-1* null mutant *C. elegans* developed a xanthine stone, suggesting additional pathways may regulate this process. To uncover such pathways, we performed a forward genetic screen for mutations that enhance the penetrance of xanthine stone formation in *xdh-1* null mutant *C. elegans*. We isolated multiple loss-of-function mutations in the gene *sulp-4* which encodes a sulfate permease homologous to human SLC26 anion exchange proteins. We demonstrated that SULP-4 acts cell-nonautonomously in the excretory cell to limit xanthine stone accumulation. Interestingly, *sulp-4* mutant phenotypes were suppressed by mutations in genes that encode for cystathionase (*cth-2*) or cysteine dioxygenase (*cdo-1*), members of the sulfur amino acid catabolism pathway required for production of sulfate, a substrate of SULP-4. We propose that sulfate accumulation caused by *sulp-4* loss of function promotes xanthine stone accumulation. We

**Data availability statement:** Whole genome sequencing data for *C. elegans* strains have been deposited at the NIH Sequence Read Archive (SRA) under accession PRJNA1208078. All other relevant data are within the paper and its Supporting information files.

**Funding:** Research reported in this publication was supported by the National Institute of General Medical Sciences of the National Institutes of Health under award number R35GM146871 (to K.W.). A.V.A. was supported by the National Science Foundation Division of Biological Infrastructure under award number 1756912. C.B. was supported by the National Institute of Childhood Health and Human Development of the National Institutes of Health under award number R25HD097633. The funders had no role in study design, data collection and analysis, decision to publish, or preparation of the manuscript.

**Competing interests:** The authors have declared that no competing interests exist.

**Abbreviations:** DIC, differential interference contrast; EMS, ethyl methanesulfonate; ICP-MS, inductively coupled plasma mass spectrometry; Moco, molybdenum cofactor; NGM, nematode growth media; pnp-1, purine nucleoside phosphorylase; qPCR, quantitative PCR; SRA, sequence read archive; SUOX-1, sulfite oxidase; WT, wild-type; XDH, xanthine dehydrogenase.

speculate that sulfate accumulation causes osmotic imbalance, creating conditions in the intestinal lumen that favor xanthine stone accumulation. Supporting this model, a mutation in *osm-8* that constitutively activates the osmotic stress response also promoted xanthine stone accumulation in an *xdh-1* mutant background. Thus, our work establishes a *C. elegans* model for human XDH deficiency and identifies the sulfate permease *sulp-4* as a critical player controlling xanthine stone accumulation.

## Introduction

Purines are an abundant and fundamental metabolite class that are essential for the generation of RNA and DNA molecules and purine nucleotides are critical energy sources (ATP) and signaling molecules (GTP). Failures in purine metabolism can lead to both common and rare diseases. For instance, oncogenic mutations activate nucleotide biosynthetic capacity in diverse cancers, promoting cancer progression [1]. Mutations in enzymes in the purine metabolic pathway cause rare inborn errors of metabolism such as Lesch–Nyhan syndrome, purine nucleoside phosphorylase deficiency, and xanthinuria [2–5]. Thus, understanding the mechanisms that impact purine homeostasis is an important goal of human health.

Xanthinuria, an inborn error of purine metabolism, is caused by inactivation of xanthine dehydrogenase (XDH), the terminal enzyme in purine catabolism that oxidizes hypoxanthine to xanthine and xanthine to uric acid [6] (Fig 1A). There are two types of human xanthinuria; type I is caused by mutations in the gene encoding the xanthine dehydrogenase enzyme and type II is caused by mutations in genes essential for the synthesis of the molybdenum cofactor, an essential prosthetic group for XDH [4,5]. Both forms of xanthinuria present with high levels of xanthine in the urine and low levels of uric acid which can result in the formation of xanthine stones in kidneys and muscles, sometimes causing renal failure. There is currently no curative therapy for xanthinuria, however, high fluid intake and low purine diets are recommended for patients [7].

Molybdenum cofactor (Moco) is an essential prosthetic group required for development in animals ranging from the nematode *Caenorhabditis elegans* to humans [4,8]. Moco is synthesized by an ancient and conserved biosynthetic pathway [9]. *C. elegans* has recently emerged as a powerful model system for studying Moco biology, and the genes that encode the Moco biosynthetic enzymes are termed *moc* in *C. elegans* for MOlybdenum Cofactor biosynthesis [8,10]. In addition to endogenous Moco synthesis, *C. elegans* can also acquire and use Moco from its bacterial diet [8,11,12]. Given its genetic tractability and the ability to manipulate animal Moco content by simple dietary manipulation, *C. elegans* is a useful model for understanding the biology of Moco and Moco-requiring enzymes, such as XDH-1 (Fig 1A).

Here, we genetically explored the formation of xanthine stones in *C. elegans,* which emerged during Moco deficiency or in *xdh-1* mutant *C. elegans,* mirroring type I and type II human xanthinuria [4,5]. Surprisingly, only a small percentage of Moco-deficient and *xdh-1* mutant *C. elegans* developed xanthine stones, suggesting

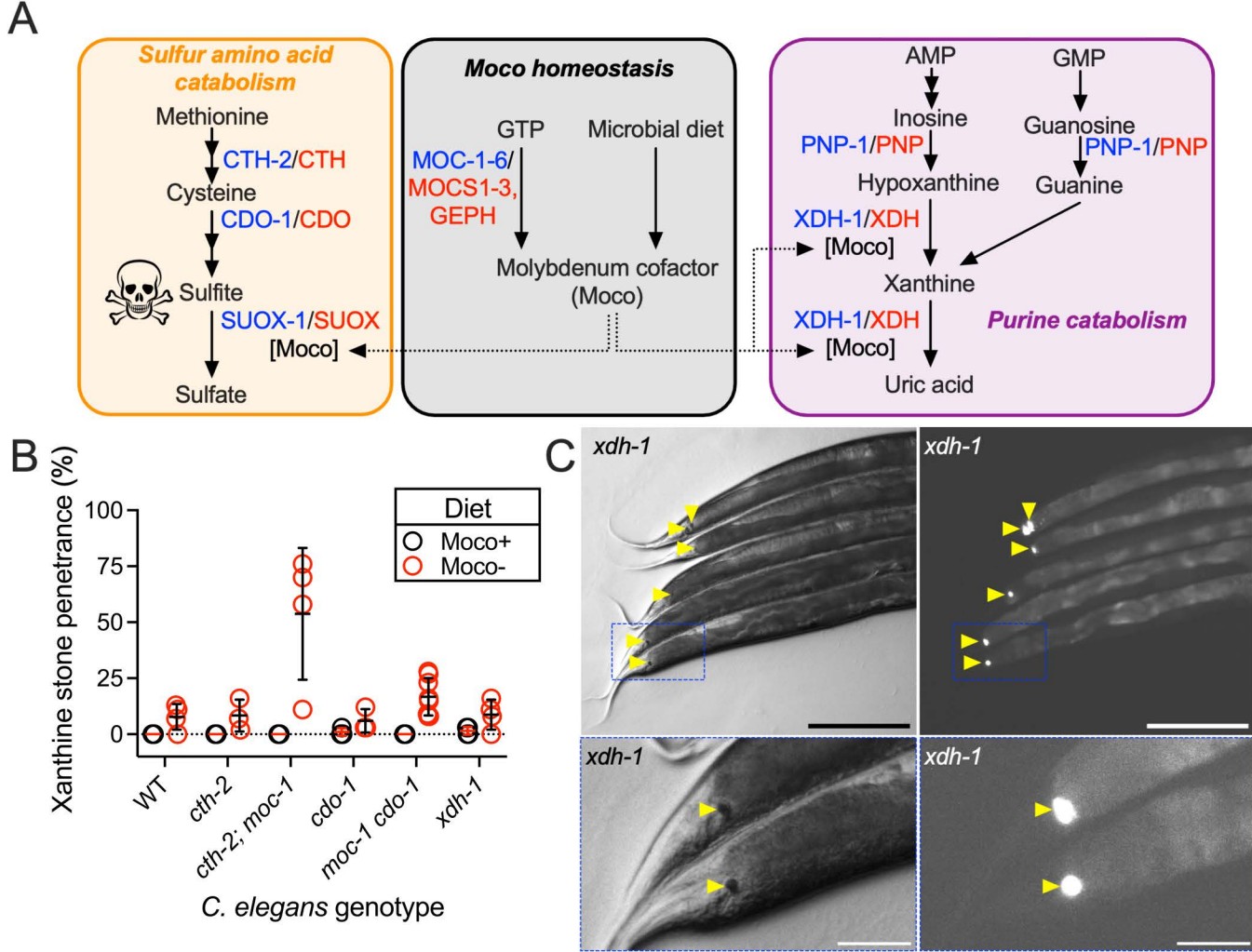

**Fig 1. Moco deficiency or loss of *xdh-1* promoted the formation of autofluorescent xanthine stones. (A)** The role of Moco in *Caenorhabditis elegans* metabolism. We highlight i) pathways for Moco synthesis and import, ii) sulfur amino acid catabolism governed by cystathionase (CTH-2/CTH), cysteine dioxygenase (CDO-1/CDO), and Moco-requiring sulfite oxidase (SUOX-1/SUOX), and iii) purine catabolism controlled by purine nucleoside phosphorylase (PNP-1/PNP) and the Moco-requiring xanthine dehydrogenase (XDH-1/XDH). *C. elegans* enzymes (blue) and their human homologs (red) are displayed. **(B)** Wild-type (WT) and mutant *C. elegans* were cultured on WT (black, Moco+) or Δ*moaA* mutant (red, Moco−) *Escherichia coli* and assessed for the formation of xanthine stones over the first 3 days of adulthood. Individual data points represent biological replicates. Mean and standard deviation are displayed. Complete information regarding sample size and individuals scored per biological replicate is found in S2 Table. **(C)** Brightfield (left) and fluorescent (right) images of the posterior of *xdh-1(ok3234)* mutant adult *C. elegans* cultured on WT *E. coli*. Xanthine stones are highlighted (yellow arrowheads). The blue box indicates the region magnified in the lower panels. Scale bar is 250 μm (top) or 50 μm (bottom).

additional parallel pathways for limiting stone formation. To identify novel regulators of xanthine stone accumulation, we performed an unbiased chemical mutagenesis screen for mutations that enhanced the penetrance of xanthine stone formation in *xdh-1* mutant animals. In this screen, we recovered five loss-of-function alleles of *sulp-4,* a gene which encodes a member of the sulfate permease family of transporters with homology to human SLC26 transporters [13]. We demonstrated that SULP-4 acts in the *C. elegans* excretory cell, analogous to the human kidney, to inhibit the accumulation of xanthine stones [14]. We further showed that *sulp-4* was required for normal development. Interestingly, we found that

phenotypes caused by *sulp-4* loss of function were suppressed by inactivating mutations in *cth-2* or *cdo-1,* genes that encode core members of the sulfur amino acid catabolism pathway (Fig 1A) [8]. We suggest that *sulp-4* loss of function causes sulfate accumulation. Disruption of *cth-2* or *cdo-1* impairs endogenous sulfate production, thereby suppressing the phenotypes observed in *sulp-4* mutant animals. Sulfate accumulation may cause an osmotic imbalance that leads to an increased rate of xanthine stone formation in the intestinal lumen. This model is supported by our observation that a mutation in *osm-8*, a key regulator of osmotic homeostasis, also promoted xanthine stone accumulation in an *xdh-1* mutant background [15]. Thus, our work establishes a *C. elegans* model for the rare genetic disease xanthinuria and identifies *sulp-4* as a potent genetic modifier of this disease pathology in *C. elegans,* likely acting through disrupted osmotic homeostasis.

## Results

### XDH-1 inactivation caused the accumulation of xanthine stones in *C. elegans*

To explore the pathology of Moco deficiency in the nematode *C. elegans,* we cultured *cth-2; moc-1* and *moc-1 cdo-1* double mutant animals on wild type (Moco replete) or Δ*moaA* mutant (Moco deficient) *Escherichia coli*. The *moc-1* mutation prevents the endogenous synthesis of Moco while the *cth-2* and *cdo-1* mutations suppress the lethality typically associated with animal Moco deficiency [8]. *cth-2* encodes cystathionase which converts cystathionine to cysteine and *cdo-1* encodes cysteine dioxygenase which oxidizes cysteine to cysteinesulfinate ultimately producing sulfites (Fig 1A). Sulfites are extremely reactive and are detoxified to sulfate by the Moco-requiring sulfite oxidase (SUOX-1) enzyme. Thus, by changing the dietary *E. coli*, we can control whether the animals have Moco. When culturing *cth-2; moc-1* double mutant animals on Moco− *E. coli,* we surprisingly observed animals that developed autofluorescent stones, typically found in the posterior of the intestine. Fifty-four percent of *cth-2; moc-1* animals fed a Moco− diet developed an autofluorescent stone while 0% of *cth-2; moc-1* animals developed an autofluorescent stone when fed a diet that provided Moco. We observed similar results for *moc-1 cdo-1* double mutant animals where 17% of animals developed an autofluorescent stone on Moco− *E. coli* and 0% developed a stone when fed wild-type (WT) *E. coli* (Fig 1B). Thus, we conclude that the formation of these autofluorescent stones is caused by Moco deficiency.

Surprisingly, we also observed the formation of autofluorescent stones in 8% of WT *C. elegans* when cultured on Moco− *E. coli.* 0% of WT animals developed an autofluorescent stone when fed Moco+ *E. coli* (Fig 1B). This result demonstrates that dietary Moco deficiency alone is sufficient to promote the formation of autofluorescent stones. This result is surprising as WT animals are still competent to produce Moco through their endogenous biosynthetic pathway. However, these data are consistent with our recent findings that the *C. elegans* diet plays a significant role in promoting Moco homeostasis [12].

Given that the development of these autofluorescent stones was dependent upon dietary Moco deficiency, we reasoned that the phenotype was likely being caused by inactivation of one of the four animal Moco-requiring enzymes (sulfite oxidase, xanthine dehydrogenase, aldehyde oxidase, and mitochondrial amidoxime reducing component) [9]. Interestingly, inactivation of the Moco-requiring enzyme xanthine dehydrogenase causes the accumulation of insoluble and fluorescent xanthine stones in organisms as diverse as plants, fruit flies, and humans [5,16–20]. We therefore hypothesized that the autofluorescent stones we observed during *C. elegans* Moco deficiency were xanthine stones. To test this, we looked for the presence of autofluorescent stones in animals carrying the *ok3234* null mutation in *xdh-1*, the *C. elegans* orthologue of xanthine dehydrogenase. In *C. elegans*, XDH-1 is expressed in the intestine, excretory cell, and neurons [21]. When we cultured *xdh-1* null mutant *C. elegans* on WT *E. coli* we indeed observed the formation of highly autofluorescent stones in 2% of animals (Fig 1B and 1C). Thus, *xdh-1* was necessary for inhibiting the formation of autofluorescent stones. Consistent with their presence in diverse models of XDH-1-deficiency, we propose that the autofluorescent stones we observe during Moco− and XDH-1-deficiency in *C. elegans* are xanthine stones.

### *sulp-4* inhibited the formation of xanthine stones

XDH-1 functions at the end of the purine catabolism pathway to oxidize hypoxanthine to xanthine and xanthine to uric acid (Fig 1A). Given the critical position of XDH-1 in purine metabolism, we were surprised that only 2% of *xdh-1* null mutant animals developed a xanthine stone. This result suggests the existence of parallel pathways for maintaining purine homeostasis. To identify additional regulators of purine metabolism, we performed an unbiased chemical mutagenesis screen for mutations that enhanced the penetrance of the xanthine stone phenotype. We mutagenized *xdh-1* mutant *C. elegans* with ethyl methanesulfonate (EMS) and cultured the newly mutagenized animals for two generations allowing newly induced mutations to become homozygous [22]. We then cloned single mutagenized F2 animals onto their own petri dish and screened for clones where we observed a high fraction of F3 progeny developing xanthine stones.

Here, we describe five new EMS-induced recessive loss-of-function mutations that caused a high penetrance of xanthine stone formation in an *xdh-1* mutant background, *rae299, rae302, rae319, rae320*, and *rae326* (see Materials and methods). These mutant alleles were prioritized because they displayed strong enhancement of xanthine stone formation and formed a complementation group, indicating they affect a single gene (see Materials and methods). To identify the causative genetic lesion in these new mutant strains, genomic DNA from all five strains was analyzed via whole-genome sequencing. Our complementation studies suggested that the mutant strains should have novel mutations in a common gene. Only one gene, *sulp-4*, was uniquely mutated in all five strains, strongly suggesting these mutations were causative for the enhanced penetrance of xanthine stone formation in the *xdh-1* mutant animals (Fig 2A and 2B and S1 Table). Among the newly isolated *sulp-4* alleles, we found 3 missense and 2 splice site mutations. Based on their recessive nature and molecular identities, we propose that these are loss-of-function alleles of *sulp-4*.

To test the hypothesis that loss of *sulp-4* function causes enhanced xanthine stone accumulation in an *xdh-1* mutant, we used CRISPR/Cas9 to engineer a new *sulp-4* deletion allele, *rae334* [23,24]. The *sulp-4(rae334)* allele is a 683 bp deletion that eliminates part of exon 1, all of exons 2 and 3, and part of exon 4 (Fig 2A). Thus, we propose that *sulp-4(rae334)* encodes a null allele. The *sulp-4(rae334)* allele strongly enhanced the penetrance of xanthine stone formation in *xdh-1* mutant *C. elegans,* phenocopying the *sulp-4* alleles isolated in our EMS screen (Fig 2B). These data demonstrate that the *sulp-4* lesions identified by whole genome sequencing cause xanthine stone accumulation in *xdh-1* mutant animals. Furthermore, these data show that *sulp-4* acts in parallel with *xdh-1* to inhibit the accumulation of xanthine stones.

The xanthine stones observed in *xdh-1; sulp-4* double mutant animals localized to the posterior of the *C. elegans* intestinal lumen, consistent with the localization of the stones observed in *xdh-1* mutant animals (Fig 2C). In addition to observing xanthine stones at a higher frequency in *xdh-1; sulp-4* mutant animals, the xanthine stones were also much larger suggesting that *sulp-4* loss of function enhances both the penetrance and expressivity of the xanthine stone phenotype in *xdh-1* mutant animals (S1 Fig). This may reflect an increased quantity of xanthine in stones displayed by *xdh-1; sulp-4* mutant animals when compared to *xdh-1* single mutant animals.

We originally observed the formation of xanthine stones during conditions of Moco deficiency; *cth-2; moc-1* double mutant animals feeding on Moco− *E. coli.* We hypothesized that the *sulp-4* mutation would also enhance the formation of xanthine stones caused by Moco deficiency. To test this, we assayed xanthine stone formation in *sulp-4* single mutant animals cultured on WT or Moco− *E. coli.* Eighty-nine percent of *sulp-4* mutant animals developed xanthine stones during dietary Moco deficiency compared to 1% of *sulp-4* mutant animals fed a Moco replete diet (S2A Fig). These data are consistent with our conclusion that *sulp-4* functions to limit the accumulation of xanthine stones caused by XDH-1 inactivation resulting from either Moco insufficiency or an *xdh-1* mutation.

### *sulp-4* promoted healthy larval and embryonic development

While culturing the *sulp-4(rae334)* mutant strain, we observed that mutant animals were sick and slow-growing. Thus, we explored the role of *sulp-4* in development and embryonic viability. To test the impact of *sulp-4* loss of function on

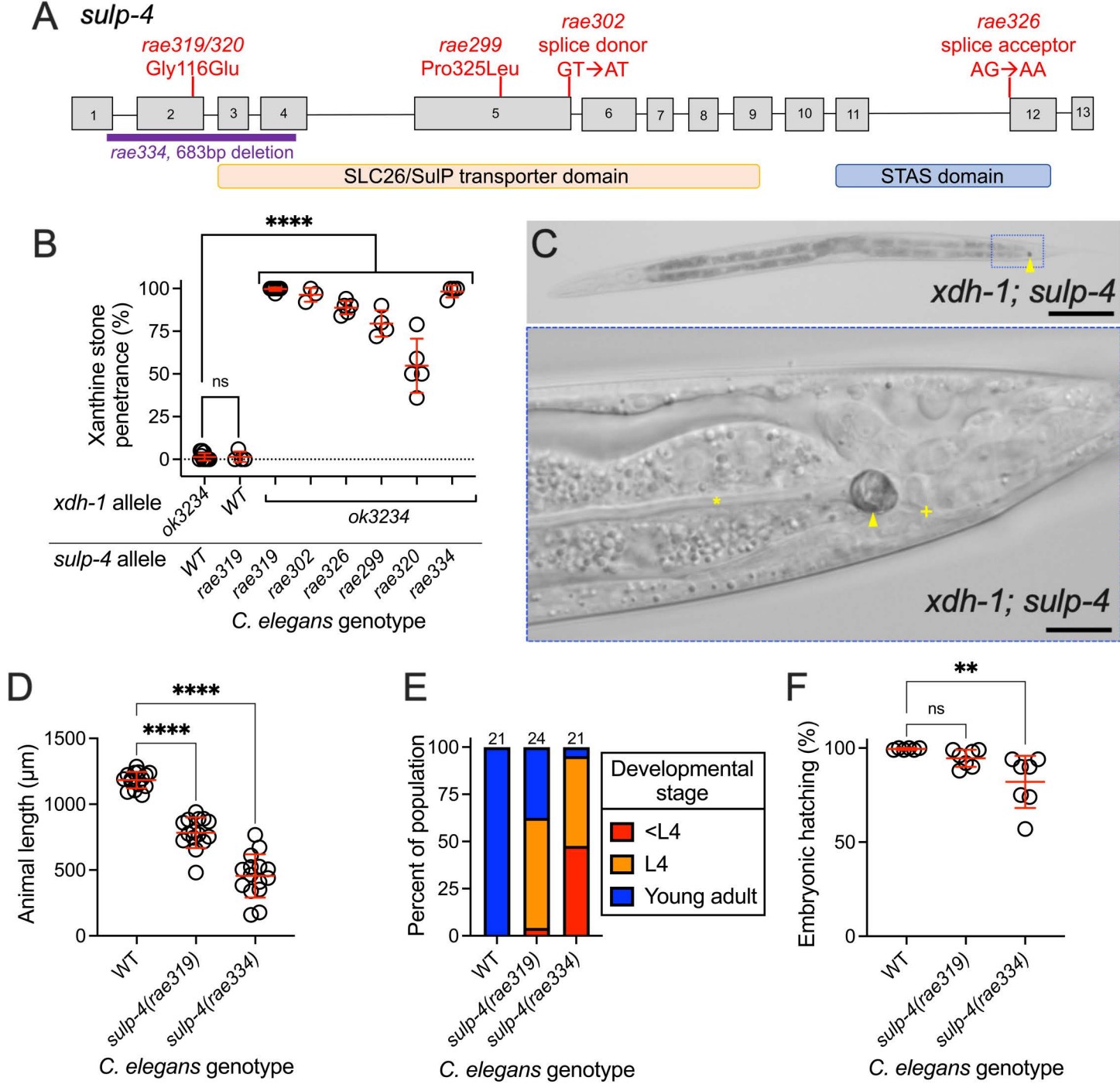

**Fig 2. Loss-of-function mutations in *sulp-4* enhanced the formation of xanthine stones during *xdh-1* deficiency. (A)** *sulp-4* locus. Gray boxes are exons and lines are introns. The regions that encode the SLC26/SulP sulfate permease domain (orange) and STAS domain (blue) are displayed. The red lines display the location of new EMS-induced lesions that enhanced the formation of xanthine stones in an *xdh-1(ok3234)* mutant background. *rae334* (purple) is a deletion allele generated using CRISPR/Cas9 technology. **(B)** Xanthine stone formation was assessed for *sulp-4(rae319)* single mutant and *xdh-1(ok3234)* mutant *Caenorhabditis elegans* with wild-type (WT) or mutant *sulp-4(rae319, rae302, rae326, rae299, rae320,* or *rae334)*. ****$p < 0.0001$, ns, $p > 0.05$, ordinary one-way ANOVA. Individual data points represent biological replicates. Mean and standard deviation are displayed. Complete information regarding sample size and individuals scored per biological replicate is found in S2 Table. **(C)** Differential interference contrast image of *xdh-1(ok3234); sulp-4(rae319) C. elegans* at the L4 stage of development. Blue box indicates the region magnified in the lower panel. The

yellow arrowhead identifies the xanthine stone. The yellow asterisk identifies the lumen of the intestine. The yellow plus sign identifies the rectum. Scale bars are 100 µm (top) and 10 µm (bottom). (**D** and **E)** WT, *sulp-4(rae319),* and *sulp-4(rae334)* mutant *C. elegans* were synchronized at the first stage of larval development and cultured for 72 hours on WT *Escherichia coli.* Animal length and developmental stage were determined. (D) Individual datapoints are displayed as are the mean and standard deviation. The sample size is 15 individuals per genotype. ****$p < 0.0001$, ordinary one-way ANOVA. (E) Sample size is the individuals assayed and is displayed above the bar graph for each genotype. (**F)** The hatching rate of newly laid WT, *sulp-4(rae319),* and *sulp-4(rae334)* mutant *C. elegans* embryos was determined. Individual data points represent biological replicates. Mean and standard deviation are displayed. Complete information regarding sample size and individuals scored per biological replicate is found in S2 Table. **$p < 0.01$ or ns, $p > 0.05$, ordinary one-way ANOVA.

developmental rate, we synchronized WT, *sulp-4(rae319),* and *sulp-4(rae334)* animals at the first stage of larval development and assayed their growth after 72 hours. We found that *sulp-4(rae334)* animals displayed a severe developmental delay compared to the wild type (Fig 2D and 2E). Interestingly, *sulp-4(rae319)* animals showed a more subtle developmental delay (Fig 2D and 2E). Thus, we propose that *sulp-4(rae334)* is a null allele while *sulp-4(rae319)* represents a hypomorph. Similarly, we observed that *sulp-4(rae334)* caused 18% embryonic lethality while *sulp-4(rae319)* caused 5% embryonic lethality. No embryonic lethality was observed for WT *C. elegans* (Fig 2F). Thus, we conclude that *rae334* and *rae319* represent an allelic series for *sulp-4* and that *sulp-4* is necessary for promoting embryonic and larval development in *C. elegans.*

### *pnp-1* was necessary for the formation of xanthine stones in *xdh-1; sulp-4* mutant animals

To further test the model that the autofluorescent stones observed in *xdh-1; sulp-4* mutant animals were composed of xanthine, we performed genetic epistasis with a null mutation in purine nucleoside phosphorylase (*pnp-1*), a gene necessary for the formation of hypoxanthine and xanthine (Fig 1A) [25]. PNP-1 is expressed and acts in the *C. elegans* intestine in addition to expression in some head neurons [25]. As previously observed, *xdh-1; sulp-4* double mutant animals displayed 98% autofluorescent stone formation while *pnp-1 xdh-1; sulp-4* triple mutant *C. elegans* displayed 3% formation of autofluorescent stones (Fig 3A). Thus, *pnp-1* was necessary for the formation of the autofluorescent stones observed in *xdh-1; sulp-4* double mutant animals. Given that *pnp-1* plays a conserved role in the formation of hypoxanthine and xanthine, these results support our model that the autofluorescent stones we observe are likely to be predominantly composed of xanthine. Although, we cannot exclude the possibility that other metabolites, such as hypoxanthine, are also present in the autofluorescent stones.

To determine if *pnp-1* acts in a genetic pathway with *sulp-4,* we tested the impact of *pnp-1* loss of function on the developmental delay displayed by *sulp-4* mutant animals. *pnp-1; sulp-4* double mutant larvae developed at a rate similar to *sulp-4* single mutant animals (Fig 3B and 3C). Importantly, *pnp-1* mutant animals displayed healthy larval development (Fig 3B and 3C). Thus, *pnp-1* was not required for the developmental delay displayed by *sulp-4* mutant *C. elegans.* We propose a genetic pathway where *pnp-1* promotes the formation of xanthine stones epistatic to the function of *xdh-1* and in parallel to the activity of *sulp-4.*

### *sulp-4*/SLC26 encodes a sulfate permease that acted in the excretory cell to promote xanthine homeostasis

*sulp-4* encodes a transmembrane transporter with homology to the SLC26 family of anion transporters in mammals [13,26]. The *C. elegans* genome encodes eight members of the SLC26 transporter family, named SULP-1 through SULP-8 [13]. We wondered if other members of the SLC26 family of transporters also played a role in limiting the formation of xanthine stones. To test this, we cultured strains with deletions in *sulp-1, sulp-2, sulp-4, sulp-5, sulp-7,* and *sulp-8* on Moco− *E. coli* and assayed the formation of xanthine stones. *sulp-3* and *sulp-6* mutant strains were not analyzed as they are inviable. Only the strain carrying the *sulp-4* mutation displayed a high penetrance of xanthine stones (88%, S2B Fig). Thus, the enhancement of xanthine stone formation is specific to loss of *sulp-4* and not a general feature of *sulp* inactivation.

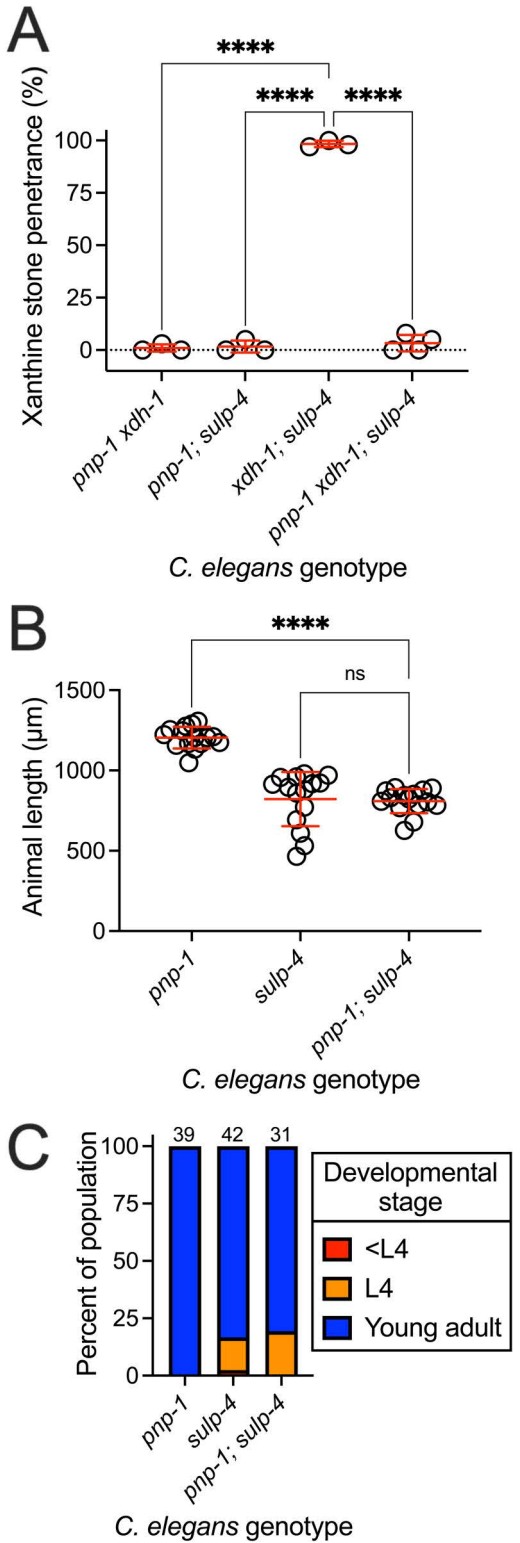

**Fig 3. *pnp-1* was required for xanthine stone formation displayed by *xdh-1; sulp-4* mutants, but not the larval delay caused by *sulp-4* loss of function. (A)** Double and triple mutant *Caenorhabditis elegans* were assessed for xanthine stone formation when cultured on wild-type (WT) *Escherichia*

*coli.* ****$p < 0.0001$, ordinary one-way ANOVA. Individual data points represent biological replicates. Mean and standard deviation are displayed. Complete information regarding sample size and individuals scored per biological replicate is found in S2 Table. **(B** and **C)** *pnp-1(jy121), sulp-4(rae319),* and *pnp-1(jy121); sulp-4(rae319)* mutant *C. elegans* were synchronized at the first stage of larval development and cultured for 72 hours on WT *E. coli.* Animal length and developmental stage were determined. (B) Individual datapoints are displayed as are the mean and standard deviation. The sample size is 15 individuals per genotype. ****$p < 0.0001$ or ns, $p > 0.05$, ordinary one-way ANOVA. (C) Sample size is the individuals assayed and is displayed above the bar graph for each genotype.

SULP-4 is expressed in the apical membrane of the *C. elegans* excretory cell, a single cell that plays roles in ionic regulation and waste elimination analogous to the mammalian renal system [13,14,27]. Studies of SULP-4 expressed in *Xenopus* oocytes demonstrate that SULP-4 is sufficient to promote the transport of sulfate and, to a lesser extent, chloride [13]. To determine the site of action of *sulp-4* with respect to the xanthine stone formation phenotype, we generated transgenic *xdh-1; sulp-4* double mutant *C. elegans* expressing a *Psulp-4::SULP-4::GFP* translational fusion transgene (plasmid was a gift from Dr. Keith Nerhke) [13]. Consistent with previous reports, we exclusively saw expression of the *Psulp-4::SULP-4::GFP* translational fusion in the excretory cell (Fig 4A and 4B). To test the functionality of the *Psulp-4::SULP-4::GFP* transgene, we performed rescue experiments with *xdh-1; sulp-4* double mutant animals expressing *Psulp-4::SULP-4::GFP* and assaying the formation of xanthine stones. *xdh-1; sulp-4* double mutant animals expressing *Psulp-4::SULP-4::GFP* did not develop xanthine stones, demonstrating functional transgenic rescue. This rescue was observed in three independently derived transgenic strains (Fig 4C). Thus, the *Psulp-4::SULP-4::GFP* transgenic fusion protein was functional, suggesting that its expression pattern faithfully represents endogenous SULP-4 localization. We conclude that SULP-4 acts in the excretory cell to negatively regulate the formation of xanthine stones. Our observation that *xdh-1; sulp-4* double mutant animals develop xanthine stones in the intestinal lumen suggests that *sulp-4* is functioning cell nonautonomously to limit the formation of xanthine stones.

To further test the role of the excretory cell in preventing the formation of xanthine stones, we used an *exc-5* mutation that causes defects in excretory cell development and morphology [28]. We reasoned a malformed excretory cell may not function efficiently and thus phenocopy *sulp-4* loss of function with respect to xanthine stone formation. Indeed, *exc-5; xdh-1* double mutant animals displayed enhanced formation of xanthine stones (S3A Fig). Although the xanthine stone penetrance of the *exc-5; xdh-1* (17%) double mutant strain was modest compared to *xdh-1; sulp-4*. Importantly, *sulp-4* loss of function did not cause cysts in the excretory cell tubules, a severe defect in excretory cell morphology caused by *exc-5* loss of function (S3B–S3D Fig). Although we cannot exclude a subtle defect in excretory cell tubule extension in *sulp-4* mutant animals. We conclude that the enhancement of xanthine stone formation caused by inactivating mutations in *sulp-4* result from loss of SULP-4 anion exchange function and not broader defects in excretory cell biology.

Furthermore, ion homeostasis plays a key role in the defecation cycle which promotes waste elimination from the *C. elegans* intestine [29]. We wondered if the enhanced formation of xanthine stones observed in *xdh-1; sulp-4* animals might result from failures in defecation, leading to xanthine accumulation in the intestinal lumen and xanthine stone formation. To test this model, we used the *aex-5(sa23)* mutation which causes defects in anterior body-wall muscle contraction (aBoc) and waste expulsion (Exp), critical aspects of the defecation cycle [30]. Interestingly, *aex-5; xdh-1* double mutant animals did not display an increased penetrance of xanthine stone formation (S3A Fig). These results suggest that defects in defecation alone are not sufficient to promote xanthine stone accumulation.

### *cth-2* and *cdo-1* were necessary for *sulp-4* mutant phenotypes

*sulp-4* inactivation caused xanthine stone formation during dietary Moco deficiency (S2A and S4 Fig), demonstrating that the enhancement of xanthine stones caused by *sulp-4* loss of function occurs even when endogenous Moco biosynthesis is functional. To test the impact of a *sulp-4* mutation on xanthine stone formation during complete Moco deficiency, we engineered *sulp-4; cdo-1 moc-1* triple mutant *C. elegans* that cannot synthesize their own Moco (caused by *moc-1*

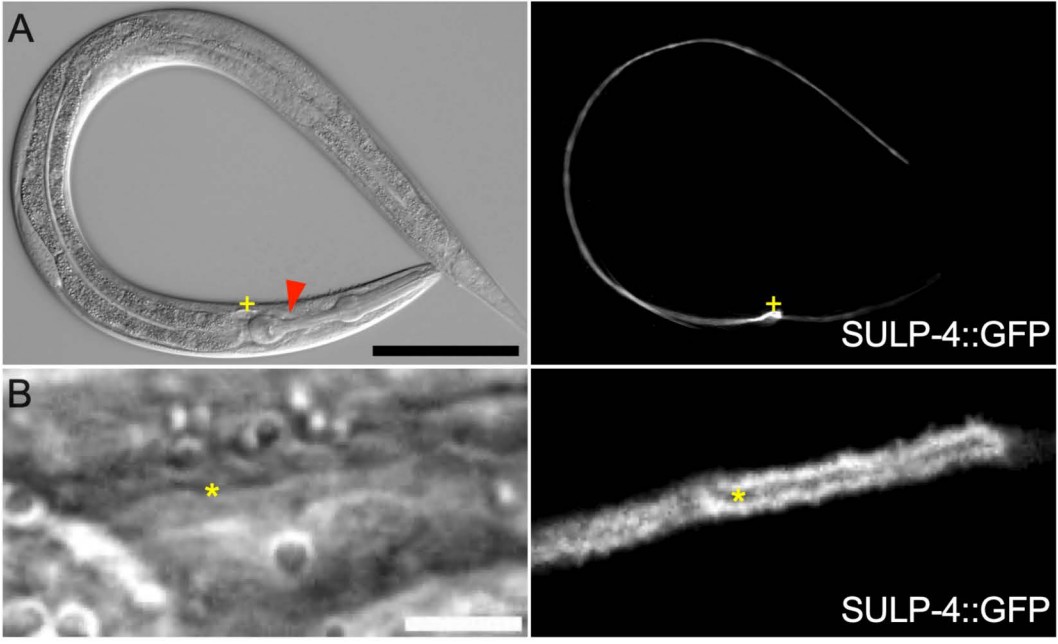

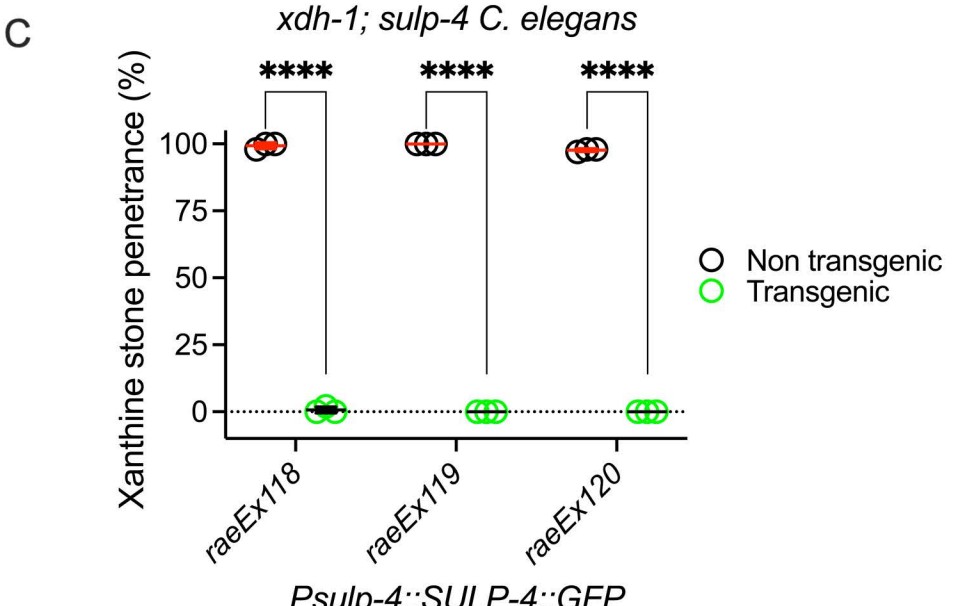

**Fig 4. *sulp-4* acted cell nonautonomously in the excretory cell to limit xanthine stone formation. (A and B)** Differential interference contrast (left) and fluorescence imaging (right) of *xdh-1(ok3234); sulp-4(rae319) Caenorhabditis elegans* expressing the *Psulp-4::SULP-4::GFP* transgene (SULP-4::GFP). The yellow plus sign identifies the cell body of the excretory cell. The red arrow indicates the region magnified in panel B. The yellow asterisk identifies the lumen of the excretory cell. Scale bars are 100 μm (A, top) and 5 μm (B, bottom). **(C)** Transgenic *xdh-1(ok3234); sulp-4(rae319) C. elegans* expressing the *Psulp-4::SULP-4::GFP* transgene (green) and their nontransgenic siblings (black) were assessed for the formation of xanthine stones. Individual data points represent biological replicates. Mean and standard deviation are displayed. ****$p < 0.0001$, multiple unpaired *t* tests. Complete information regarding sample size and individuals scored per biological replicate is found in S2 Table.

mutation) and are viable during Moco deficiency (caused by *cdo-1* suppressor mutation) (Fig 1A). When cultured on Moco− *E. coli*, *sulp-4; cdo-1 moc-1* triple mutants are completely Moco deficient yet only displayed 9% penetrance of xanthine stones (S4 Fig). Similarly, *sulp-4; cdo-1* double mutant animals cultured on Moco− *E. coli* displayed 0% penetrance of xanthine stones, dramatically reduced compared to the 86% penetrance displayed by *sulp-4* mutant animals cultured on Moco− *E. coli* (S4 Fig). These results were surprising and suggest that *cdo-1* is necessary for the formation of xanthine stones caused by a *sulp-4* mutation and Moco deficiency.

To further test the impact of *cdo-1* on the formation of xanthine stones, we engineered *xdh-1; sulp-4; cdo-1* triple mutant *C. elegans*. Surprisingly, *xdh-1; sulp-4; cdo-1* triple mutant animals displayed a 5% xanthine stone penetrance, dramatically reduced when compared to the 98% penetrance displayed by *xdh-1; sulp-4* double mutant animals (Fig 5A). Thus, *cdo-1* was necessary for the formation of xanthine stones displayed by *xdh-1; sulp-4* double mutant animals. Interestingly, we still observe a low penetrance of xanthine stones in *xdh-1; sulp-4; cdo-1* triple mutant *C. elegans* suggesting that *cdo-1* activity is not absolutely required for the formation of xanthine stones but only required for the xanthine stone enhancement caused by *sulp-4* loss of function.

*cdo-1* encodes the *C. elegans* cysteine dioxygenase, a critical enzyme in the sulfur amino acid catabolism pathway that breaks down excess cysteine and methionine (Fig 1A) [8,31,32]. To determine if the impact of *cdo-1* on the enhanced xanthine stone formation caused by *sulp-4* inactivation was a result of impaired sulfur amino acid catabolism, we used a *cth-2* mutation which eliminates the activity of *C. elegans* cystathionase (Fig 1A). Consistent with our results with *cdo-1* loss of function, we found that *cth-2; xdh-1; sulp-4* triple mutant animals also displayed a low 19% penetrance of xanthine stones (Fig 5A). Taken together, these genetic data suggest that sulfur amino acid catabolism is required for the enhancement of xanthine stone accumulation caused by loss of *sulp-4* function.

Given that *sulp-4* encodes a sulfate permease, we hypothesized that sulfate accumulation during *sulp-4* loss of function was promoting the formation of xanthine stones. Furthermore, we hypothesized that this sulfate was produced endogenously via CTH-2 and CDO-1. To further test this model, we measured sulfur content in large cultures of WT and *sulp-4* mutant *C. elegans* using inductively coupled plasma mass spectrometry (ICP-MS). We did not observe any difference in sulfur content between WT and *sulp-4* mutant animals, demonstrating that *sulp-4* mutant *C. elegans* do not accumulate sulfur (S5B Fig). Similarly, we saw no difference in K, Fe, Zn, Mn, Cu, or Mo content between wild-type and *sulp-4* mutant animals (S5 Fig). A key limitation of this method is that ICP-MS does not distinguish between sulfur found in different metabolites, such as sulfate, sulfite, methionine, cysteine, etc. Thus, it is plausible that *sulp-4* mutant animals accumulate sulfate with a corresponding decrease in another sulfur-containing metabolite. Alternatively, sulfate sulfur may only account for a small fraction of the animal sulfur economy. Thus, physiologically significant changes in sulfate may be undetectable given the high background of sulfur found in more common metabolites like cysteine or methionine.

To further test the relationship between sulfur amino acid catabolism and xanthine stone formation, we supplemented *xdh-1; sulp-4; cdo-1* or *cth-2; xdh-1; sulp-4* triple mutant *C. elegans* with the sulfur-containing amino acid cysteine and assessed the impact on xanthine stone formation. Cysteine supplementation is informative as it is the product of CTH-2 and the substrate for CDO-1 (Fig 1A). We found that cysteine promoted xanthine stone formation in *cth-2; xdh-1; sulp-4* animals, bypassing the requirement for *cth-2*. Alternatively, cysteine did not promote xanthine stones in *xdh-1; sulp-4; cdo-1* mutant animals, demonstrating that *cdo-1* was necessary for cysteine to promote xanthine stone formation (Fig 5B). This result supports our model that endogenous sulfur amino acid catabolism promotes xanthine stone formation. More specifically, these data demonstrate that a cysteine-derived metabolite downstream of CDO-1 is acting to promote xanthine stones. This is consistent with our model that sulfate accumulation during *sulp-4* loss of function promotes xanthine stone accumulation.

To test whether mutations in *cth-2* or *cdo-1* would suppress the defects in larval and embryonic development displayed by *sulp-4* mutant animals, we assayed larval and embryonic development in *cth-2; sulp-4* and *sulp-4; cdo-1* double mutant animals and compared to *sulp-4* single mutant animals. Consistent with their suppression of xanthine stone formation,

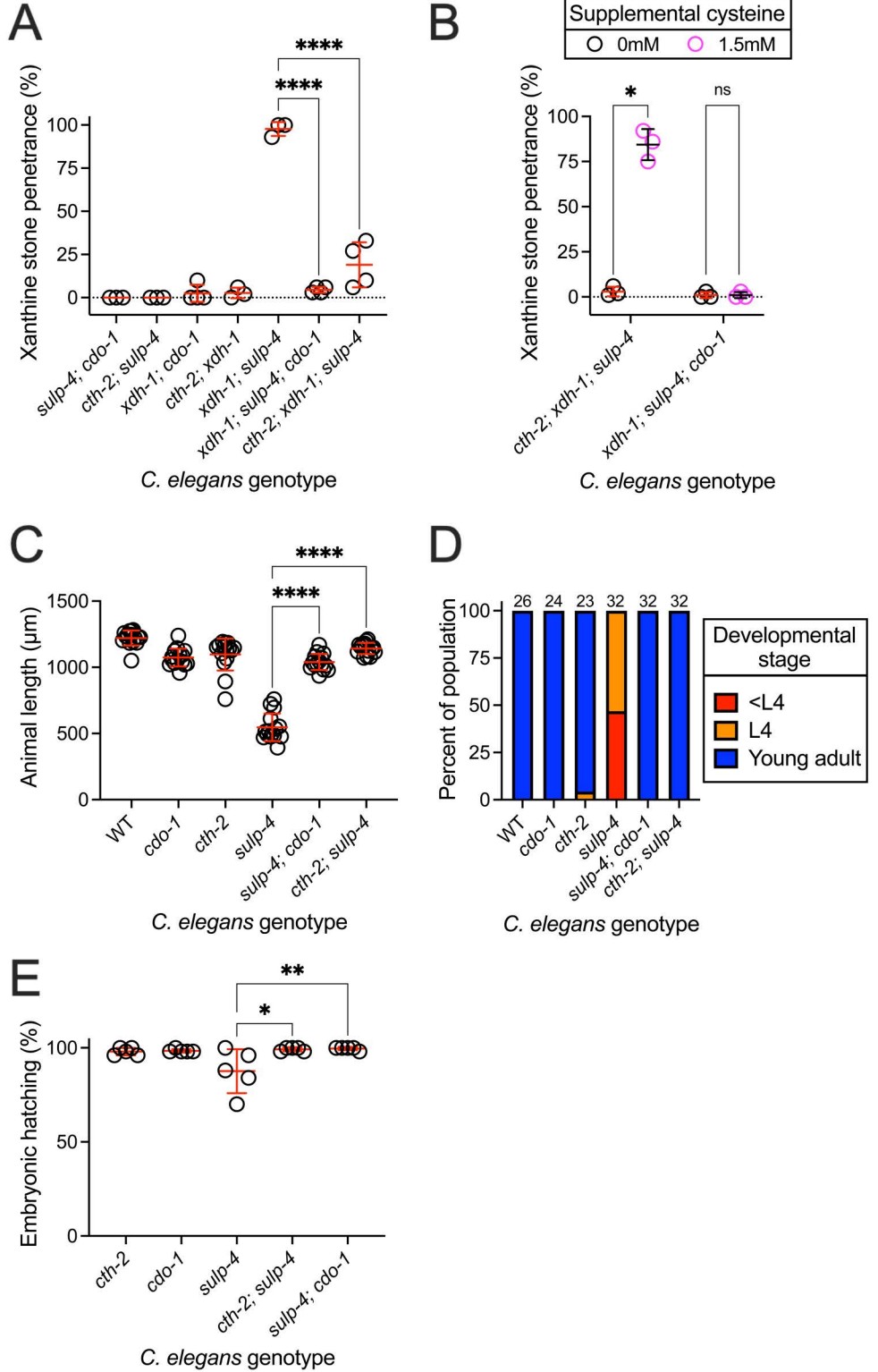

**Fig 5. *cth-2* and *cdo-1* were required for phenotypes caused by *sulp-4* loss of function. (A)** Double and triple mutant *Caenorhabditis elegans* were assessed for xanthine stone formation when cultured on wild-type (WT) *Escherichia coli* over the first 3 days of adulthood. Individual data points represent biological replicates. Mean and standard deviation are displayed. ****$p < 0.0001$, ordinary one-way ANOVA. Complete information regarding sample

size and individuals scored per biological replicate is found in S2 Table. **(B)** Triple mutant *C. elegans* exposed to 0 or 1.5 mM supplemental cysteine were assessed for xanthine stone formation when cultured on WT *E. coli* over the first 2 days of adulthood. Mean and standard deviation are displayed. $*p < 0.05$ and ns, $p > 0.05$, multiple unpaired *t* tests. Complete information regarding sample size and individuals scored per biological replicate is found in S2 Table. **(C and D)** WT and mutant *C. elegans* were synchronized at the first stage of larval development and cultured for 72 hours on WT *E. coli*. Animal length and developmental stage were determined. (C) Individual datapoints are displayed as are the mean and standard deviation. The sample size is 15 individuals per genotype. $****p < 0.0001$, ordinary one-way ANOVA. (D) Sample size is the individuals assayed and is displayed above the bar graph for each genotype. **(E)** The hatching rate of newly laid single and double mutant *C. elegans* embryos was determined. Individual data points represent biological replicates. Mean and standard deviation are displayed. $**p < 0.01$ or $*p < 0.05$, ordinary one-way ANOVA. Complete information regarding sample size and individuals scored per biological replicate is found in S2 Table. The *sulp-4(rae334)* allele was used to generate the data in Fig 5.

*cth-2* or *cdo-1* mutations suppressed the developmental delay and embryonic lethality caused by a *sulp-4* mutation (Fig 5C–5E). Thus, we conclude that *cth-2* and *cdo-1* are broadly required for phenotypes caused by *sulp-4* loss of function.

### Loss of *osm-8* promoted xanthine stone accumulation in *xdh-1* mutant animals, linking osmotic homeostasis and xanthine stone formation

What is the mechanism by which sulfates, derived from CTH-2 and CDO-1, might promote xanthine stone formation when *sulp-4* is inactivated? Given that *sulp-4* encodes a sulfate permease that functions in the *C. elegans* excretory cell, we hypothesized that *sulp-4* mutant animals may be experiencing osmotic imbalance driven by excess sulfate. Supporting this model, *sulp-4* mutant animals accumulated 25% more sodium than their WT counterparts as measured by ICP-MS (Fig 6A). These data demonstrate that *sulp-4* mutant animals are experiencing osmotic imbalance.

To test the model that osmotic imbalance promotes the formation of xanthine stones, we challenged *xdh-1* mutant animals with supplemental NaCl. Interestingly, *xdh-1* mutant animals were extremely sensitive to high environmental NaCl when compared to wild type, a phenotype also displayed by rats lacking *xdh-1* (Fig 6B) [33]. These data indicate a conserved role for *xdh-1* in promoting hyperosmotic stress tolerance. However, the sensitivity of *xdh-1* mutant *C. elegans* to NaCl limited our ability to directly test whether hyperosmotic stress promotes xanthine stone formation in an *xdh-1* mutant background.

To genetically test whether osmotic imbalance may promote xanthine stones, we used a loss-of-function mutation in *osm-8* that constitutively activates the hyperosmotic stress response in the absence of environmental osmotic stress [15,34]. *osm-8* encodes a mucin-like protein and is expressed in the *C. elegans* hypodermis [15]. Interestingly, *osm-8; xdh-1* double mutant *C. elegans* developed xanthine stones like *xdh-1; sulp-4* double mutant animals (Fig 6C). Although, the xanthine stones observed in *osm-8; xdh-1* double mutant animals were not as large as those observed in *xdh-1; sulp-4* animals (S1 Fig). Furthermore, like *xdh-1; sulp-4* double mutant *C. elegans*, *cdo-1* was necessary for the formation of xanthine stones in *osm-8; xdh-1* double mutant animals (Fig 6C). However, it should be noted that *cdo-1* loss of function incompletely suppressed the formation of xanthine stones in *osm-8; xdh-1* mutant animals, suggesting additional pathways downstream of *osm-8* that govern xanthine stone formation. These data suggest that altered osmotic homeostasis, driven by *osm-8* loss of function, is sufficient to cause xanthine stone accumulation when XDH-1 is inactive.

We wondered if *sulp-4* and *osm-8* mutant animals display additional overlapping phenotypes. *osm-8* mutant *C. elegans* are resistant to high NaCl stress and display increased transcription of the hypertonic stress response which includes the key regulators of osmolyte accumulation, *gpdh-1* and *hmit-1.1* [15,34]. *gpdh-1* encodes glycerol 3-phosphate dehydrogenase and is required to produce glycerol, a critical osmolyte [35,36]. *hmit-1.1* encodes an H$^+$/*myo*-inositol transporter and functions to import *myo*-inositol, another important osmolyte [37]. Distinct from *osm-8* mutant animals, *sulp-4* mutant *C. elegans* were not resistant to high NaCl (Fig 6B). Furthermore, *sulp-4* mutant animals did not significantly accumulate *hmit-1.1* mRNA and modestly accumulated *gpdh-1* mRNA when compared to the wild type (S6A and S6B Fig). These data suggest that *sulp-4* inactivation does not robustly activate transcription of all canonical hypertonic stress response genes. Interestingly, both *sulp-4* and *osm-8* mutant animals accumulated *cdo-1* mRNA when compared to the wild type (Fig 6D), consistent with previous

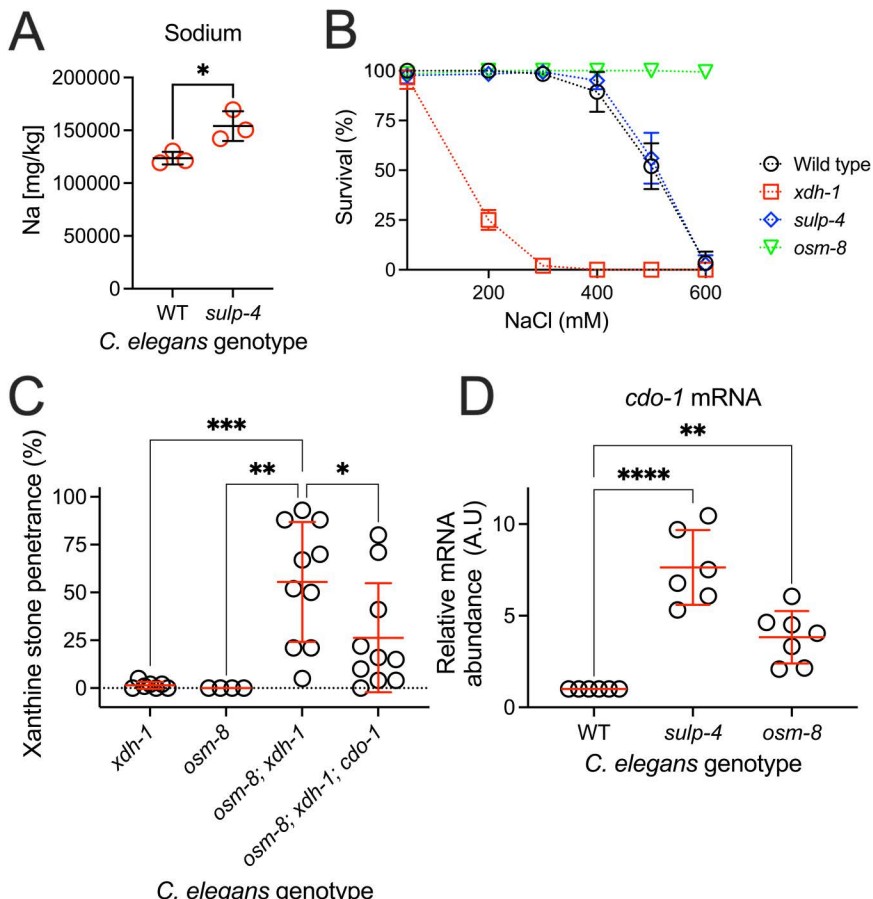

**Fig 6. Loss of *osm-8*, which activates the osmotic stress response, enhanced xanthine stone accumulation in *xdh-1* mutant *Caenorhabditis elegans*.** (A) Sodium content was determined for synchronized wild-type (WT) and *sulp-4(rae319)* young adult *C. elegans* using ICP-MS. Individual data points represent biological replicates. Mean and standard deviation are displayed. **p < 0.05*, *t* test. **(B)** Survival of WT, *sulp-4(rae319)*, *xdh-1(ok3234)*, or *osm-8(n1518)* young adult *C. elegans* was determined after 24-hour exposure to nematode growth media containing various concentrations of NaCl. Mean and standard deviation of 3 biological replicates are displayed. Complete information regarding sample size and individuals scored per biological replicate is found in S2 Table. **(C)** Mutant *C. elegans* were assessed for xanthine stone formation when cultured on WT *Escherichia coli*. Individual data points represent biological replicates. Mean and standard deviation are displayed. ****p < 0.001*, ***p < 0.01*, **p < 0.05*, ordinary one-way ANOVA. Complete information regarding sample size and individuals scored per biological replicate is found in S2 Table. **(D)** Relative mRNA levels of *cdo-1* are displayed for RNA isolated from WT, *sulp-4(rae319)*, and *osm-8(n1518)* young adult *C. elegans*. Relative mRNA abundance was determined via the delta-delta $C_T$ method. All transcripts are normalized to *act-1*. Relative mRNA abundance for each transcript was set to one in the wild type. *****p < 0.0001*, ***p < 0.01*, ordinary one-way ANOVA.

studies in *C. elegans* and mammals that demonstrate *cdo-1* induction by osmotic imbalance (high NaCl exposure or *osm-8* inactivation) [34,38]. Taken together, these data highlight overlapping but distinct phenotypic profiles for *sulp-4* and *osm-8* mutant *C. elegans* and elucidate a positive feedback loop connecting *sulp-4*, osmotic imbalance, and *cdo-1* activation.

In addition to its role in cysteine catabolism, CDO-1 is also essential for synthesis of the osmolyte taurine [32,38]. Given that *cdo-1* was necessary for phenotypes caused by *sulp-4* inactivation, we wondered whether inactivating mutations in other genes involved in osmolyte accumulation may also suppress *sulp-4* mutant phenotypes. To test this, we engineered *gpdh-1; sulp-4* and *hmit-1.1; sulp-4* double mutant strains of *C. elegans* and assayed for the formation of xanthine stones when animals were cultured on a Moco− diet. Unlike *cdo-1*, neither *gpdh-1* nor *hmit-1.1* was necessary for the formation of xanthine stones caused by a *sulp-4* mutation (S6C Fig). These data suggest that the suppression of *sulp-4* mutant

phenotypes by *cdo-1* loss-of-function is specific and not a general feature of inactivating mutations in genes involved in the osmotic stress response. However, it is important to note that the *C. elegans* genome encodes homologs of both *gpdh-1* and *hmit-1.1* which may act redundantly, complicating the mechanistic interpretation of these genetic data [35].

We propose the model that in healthy WT *C. elegans,* cysteine and methionine are being broken down by the sulfur amino acid catabolism pathway (CTH-2/CDO-1), maintaining sulfur homeostasis. This results in the production of sulfate which is exported into the environment via the action of SULP-4 in the excretory cell. However, when *sulp-4* is inactive, sulfate cannot be safely excreted and accumulates. This issue is exacerbated by an induction of *cdo-1* mRNA during *sulp-4* loss of function. This creates a maladaptive positive feedback loop that we anticipate would further increase sulfate production. We speculate that sulfate accumulation then promotes osmotic imbalance which causes embryonic lethality, impaired larval development, and an increased propensity to form xanthine stones when *xdh-1* is inactive (Fig 7).

## Discussion

### Modeling xanthinuria in *C. elegans*

Human xanthinuria was originally described in 1954, and presents with high urinary xanthine, low uric acid in serum and urine, the formation of xanthine stones, and, in some cases, renal failure [5,39]. Still, there are no curative treatments for xanthinuria or the formation of xanthine stones. The current recommendation for patients is a high fluid intake and low purine diet [7]. Thus, understanding the cellular mechanisms that regulate the pathology associated with xanthinuria is an important goal.

Animal models, such as *C. elegans,* are powerful tools for exploring the pathology of rare inborn errors of metabolism, including NGLY1 deficiency, Moco deficiency, Friedrich's ataxia, and many others [8,40,41]. Here, we used *C. elegans* to model human type I and type II xanthinuria. We employed genetic strategies to inhibit XDH-1 activity by mutating the *xdh-1* gene (type I) or limiting animal Moco (type II) [4,5]. Both manipulations recapitulated a critical feature of human XDH deficiency, the formation of insoluble xanthine stones.

Interestingly, xanthine stones are highly autofluorescent and visible with a standard fluorescence microscope, a phenotype that has been previously characterized in the model plant *Arabidopsis thaliana* [18]. Given the transparent nature of *C. elegans,* this phenotype empowers genetic analyses of xanthine stone accumulation and, by proxy, purine biology. Here, we used the power of *C. elegans* genetics in combination with this simple phenotype to identify and characterize regulators of purine homeostasis.

### Defining genetic regulators of xanthine stone formation

We sought to define genetic regulators of the formation of xanthine stones. Given the established purine catabolism pathway (Fig 1A), we used a hypothesis-driven approach to define genes that regulate the formation of xanthine stones. Purine nucleoside phosphorylase (PNP-1/PNP) was a lead candidate given its biochemical requirement for the formation of xanthine. Indeed, we demonstrated that *pnp-1* was necessary for the formation of xanthine stones in our *C. elegans* mutant animals: *pnp-1* loss of function suppressed the formation of xanthine stones in an *xdh-1; sulp-4* double mutant background. These results suggest that inhibiting the activity of PNP may be a therapeutic strategy for limiting the accumulation of xanthine and xanthine stones in patients suffering from xanthinuria. However, this treatment strategy may be fraught, given the consequences of PNP inactivation. Human patients with PNP deficiency display impaired T-cell immunity [3]. In fact, a potent PNP inhibitor has been developed, and induces apoptosis of B- and T-lymphocytes [42,43]. Thus, the potential benefits of PNP inhibition in the treatment of human xanthinuria patients would need to be evaluated and weighed against the negative impacts on the immune system.

To identify new and unexpected regulators of purine homeostasis, we employed an unbiased genetic approach. In a forward genetic screen, we identified *sulp-4* as a potent modifier of xanthine stone formation. Loss-of-function mutations in *sulp-4* dramatically enhanced the penetrance and expressivity of xanthine stone formation in our *C. elegans* models of xanthinuria. Interestingly, we also found that *sulp-4* was necessary for promoting normal larval and embryonic

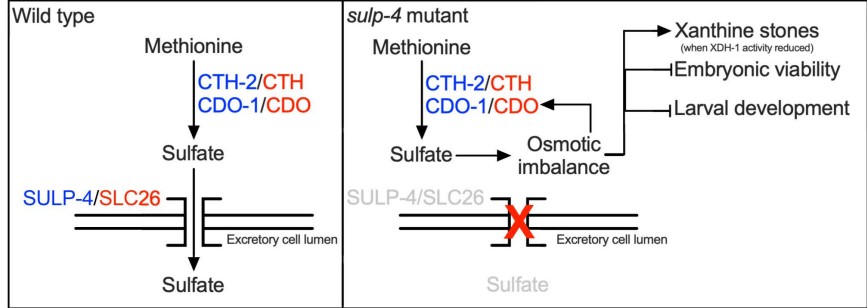

**Fig 7. SULP-4 promotes sulfate homeostasis, maintaining osmotic balance in *C. elegans*.** In wild-type *C. elegans*, sulfur amino acid catabolism gives rise to sulfate which is maintained at homeostatic levels by SULP-4-mediated exchange with the environment via the excretory cell. During *sulp-4* loss of function, CTH-2/CDO-1-derived sulfates accumulate causing osmotic imbalance. This osmotic imbalance promotes a maladaptive positive feedback loop promoting additional *cdo-1* mRNA accumulation. This cascade culminates in embryonic lethality, altered larval development, and a propensity to form xanthine stones when XDH-1 activity is compromised.

development. These genetic data demonstrate that *sulp-4* promotes healthy development and acts in a parallel pathway to *xdh-1* to limit the accumulation of xanthine stones.

*sulp-4* was previously demonstrated to encode a sulfate permease that localizes to the apical membrane of the excretory cell, which we also observed [13]. Importantly, we demonstrated that *Psulp-4::SULP-4::GFP* rescued the formation of xanthine stones displayed by an *xdh-1; sulp-4* double mutant animal. This functional rescue is strong evidence that *sulp-4* is acting in the excretory cell to limit the formation of xanthine stones. Interestingly, xanthine stones accumulate in the lumen of the *C. elegans* intestine while *sulp-4* acts in the excretory cell. Thus, we conclude that *sulp-4* acts cell-nonautonomously to limit xanthine stone accumulation. These data establish a new intersection between SULP-4, the excretory cell, and purine homeostasis.

## Linking osmotic homeostasis and xanthine stone formation

We next sought to understand the nature of the genetic interaction between *sulp-4* and *xdh-1* with respect to xanthine stone formation. Given that *sulp-4* encodes a sulfate transporter that acts cell-nonautonomously in the excretory cell, we hypothesized that *sulp-4* mutations may cause sulfate retention and disturb osmotic balance in *C. elegans* systemically [13]. Indeed, we found that *sulp-4* mutant *C. elegans* accumulated sodium, demonstrating osmotic imbalance [44,45]. Importantly, we do not know the site of excess sodium accumulation (cellular versus extracellular) in *sulp-4* mutant *C. elegans*. We propose the model that osmotic imbalance in *sulp-4* mutant animals is driven by a failure to excrete endogenously produced sulfates. This model is supported by our genetic demonstration that *cth-2* or *cdo-1*, genes that encode enzymes necessary for sulfate production, were necessary for xanthine stone formation in *xdh-1; sulp-4* mutant animals. We speculate that osmotic imbalance driven by sulfate retention may promote water reabsorption causing dehydration in the intestinal lumen, increasing xanthine concentration and stone formation. This model is reinforced by our observation that activation of the hyperosmotic stress response through a distinct genetic perturbation, *osm-8* loss of function, also promoted the formation of xanthine stones in an *xdh-1* mutant background [15]. In the case of *osm-8* inactivation, we speculate that glycerol accumulation would drive water reabsorption and dehydration in the intestinal lumen. Again, this dehydration would increase xanthine concentration, promoting stone formation.

## Intersection between sulfur amino acid metabolism and Moco-dependent metabolism

Previous studies in *C. elegans* identified *cth-2* and *cdo-1* loss-of-function mutations as suppressors of the lethality associated with Moco deficiency and deficiency of the Moco-requiring enzyme sulfite oxidase [8]. Loss of sulfite oxidase is lethal in *C. elegans* and humans due to the accumulation of its reactive and toxic substrate, sulfite (Fig 1A) [8,46]. *cth-2* and

*cdo-1* inactivation limit the accumulation of sulfites, suppressing the lethality caused by Moco or sulfite oxidase deficiency. Is it a coincidence that loss of *cth-2* or *cdo-1* suppress phenotypes associated with two distinct Moco-requiring enzymes: i) lethality displayed by *suox-1* null mutant animals and ii) xanthine stone formation of *xdh-1; sulp-4* double mutant animals? Given that the substrates and products of these two Moco-dependent enzymes are distinct, it is peculiar that loss-of-function phenotypes of both are modulated by these common genetic factors. Future studies are required to tease apart the potential molecular intersections between CTH-2/CDO-1 and these Moco-dependent pathways.

An additional layer of complexity is added when considering the regulation of *cdo-1.* Here, we reinforce previous observations that *cdo-1* is activated in response to hyperosmotic stress [34,38]. *sulp-4* and *osm-8* mutant *C. elegans* both accumulated *cdo-1* mRNA. Yet, CDO-1/CDO levels and activity are also modified by dietary sulfur amino acid content. This regulation includes both transcriptional and post-translational control of cysteine dioxygenase [31,47–51]. Given that SULP-4 mediates sulfate transport and sulfate is a metabolic product of sulfur amino acid catabolism governed by CTH-2/CDO-1/SUOX-1, sulfate seems like a potential metabolic intersection between these seemingly disparate Moco-dependent pathways. Whether there is any overlap between the mechanisms of CDO-1/CDO regulation downstream of hyperosmotic stress and high sulfur amino acid content remains to be studied.

### SULP-4 homologs are implicated in calcium oxalate stone formation

Our genetic studies demonstrate that SULP-4 plays a role in limiting the formation of xanthine stones in *C. elegans* models of xanthinuria. SULP-4 is homologous to human SLC26 proteins which have been previously linked to metabolic stone formation. Understanding the connection between SLC26 proteins, like SULP-4, and the formation of metabolic stones is important as 1 out of every 10 individuals in the United States will develop a kidney stone over the course of their lifetime [52]. For instance, mouse SLC26A6 limits urolithiasis, paralleling our results with *C. elegans* SULP-4. Specifically, SLC26A6 null mutant mice develop a high incidence of calcium oxalate stones in the bladder [53]. Similarly, a dominant negative mutation in SLC26A6 and loss-of-function mutations in SLC26A1 are proposed to cause calcium oxalate nephrolithiasis in humans [54,55]. In contrast, the *Drosophila melanogaster* homolog of human SLC26A5/6, dPrestin, promotes the formation of calcium oxalate stones in the Malpighian tubules in a dietarily-induced model of calcium oxalate nephrolithiasis. Under a diet of high oxalate, *Drosophila* develop calcium oxalate stones whose formation is dampened by RNAi knockdown of dPrestin [56]. With respect to calcium oxalate stones, the mechanism of stone formation is believed to relate directly to the role of the SLC26 family members in oxalate transport. Loss of SLC26 proteins results in higher or lower concentrations of oxalate in a given space, altering the likelihood of stone formation. We propose that xanthine is the critical component of the stones that form in *xdh-1; sulp-4* mutant animals, not oxalate. In heterologous transport assays, oxalate was not meaningfully transported by SULP-4 [13]. However, it is possible that *sulp-4* mutations may enhance the formation of additional metabolic stones given a specific genetic or environmental perturbation. This hypothesis remains to be tested. Regardless, we think it is very intriguing that SLC26 homologs in mouse, humans, flies, and worms have all been shown to play roles in metabolic stone formation and suggest the potential for a fundamental mechanism underlying these discrete observations.

## Materials and methods

### General methods and strains

*C. elegans* were cultured using established protocols [22]. Briefly, animals were cultured at 20 °C on nematode growth media (NGM) seeded with WT *E. coli* (OP50) unless otherwise noted. The WT strain of *C. elegans* was Bristol N2. Additional *E. coli* strains used in this work were BW25113 (Wild type, Moco+) and JW0764-2 (Δ*moaA753::kan,* Moco−) [57].

C. elegans mutant and transgenic strains used in this work are listed here. When previously published, sources of strains are referenced. Unless a reference is provided, all strains were generated in this study.

**Non-transgenic strains.**

N2, wild type [22]

GR2257, cth-2(mg599) II [8]

GR2259, cth-2(mg599) II; moc-1(ok366) X [8]

GR2260, cdo-1(mg622) [8]

GR2261, cdo-1(mg622) moc-1(ok366) X [8]

USD869, xdh-1(ok3234) IV (Outcrossed 2X)

USD1033, sulp-4(rae319) V (Outcrossed 4X)

USD1037, sulp-4(rae319) V; cdo-1(mg622) moc-1(ok366) X

USD1038, sulp-4(rae319) V; cdo-1(mg622) X

USD1055, xdh-1(ok3234) IV; sulp-4(rae334) V

USD1091, sulp-4(rae334) V; cdo-1(mg622) X

USD1103, cth-2(mg599) II; sulp-4(rae334) V

USD1105, cth-2(mg599) II; xdh-1(ok3234) IV; sulp-4(rae334) V

USD1146, xdh-1(ok3234) IV; sulp-4(rae334) V; cdo-1(mg622) X

USD1154, xdh-1(ok3234) IV; cdo-1(mg622) X

USD1163, pnp-1(jy121) IV (Outcrossed 1X) [25]

USD1170, cth-2(mg599) II; xdh-1(ok3234) IV

USD1174, pnp-1(jy121) xdh-1(ok3234) IV

USD1198, pnp-1(jy121) IV; sulp-4(rae319) V

USD1215, pnp-1(jy121) xdh-1(ok3234) IV; sulp-4(rae319) V

USD1230, aex-5(sa23) I; xdh-1(ok3234) IV

USD1269, exc-5(rh232) xdh-1(ok3234) IV

USD1308, osm-8(n1518) II; xdh-1(ok3234) IV

USD1310, gpdh-1(ok1558) I (Outcrossed 4X)

USD1312, hmit-1.1(ok2923) V (Outcrossed 4X)

USD1322, gpdh-1(ok1558) I; sulp-4(rae319) V

USD1324, sulp-4(rae319) hmit-1.1(ok2923) V

USD1327, osm-8(n1518) II; xdh-1(ok3234) IV; cdo-1(mg622) X

JT23, *aex-5(sa23) I [*30*]*

NJ731, *exc-5(rh232) IV [*28]

MT3571, *osm-8(n1518) II [*15]

RB1082, *sulp-5(ok1048) V*

RB1366, *sulp-2(ok1551) X*

RB1369, *sulp-2(ok1554) X*

RB1436, *sulp-1(ok1639) I*

RB2134, *sulp-8(ok2842) V*

FX08263, *sulp-5(tm8264) X*

VC3021, *sulp-7(ok3751) X*

VC3045, *sulp-7(ok3752) X*

**Transgenic strains.**

USD1060, *xdh-1(ok3234) IV; sulp-4(rae319) V; raeEx118*

USD1061, *xdh-1(ok3234) IV; sulp-4(rae319) V; raeEx119*

USD1062, *xdh-1(ok3234) IV; sulp-4(rae319) V; raeEx120*

USD1251, *qpIs11 I; sulp-4(rae319) V*

USD1277, *qpIs11 I; exc-5(rh232) IV*

BK36, *qpIs11 I; unc-119(ed3) III [*58]

**EMS-derived strains.**

USD962*, *xdh-1(ok3234) IV; sulp-4(rae299) V*

USD1007, *xdh-1(ok3234) IV; sulp-4(rae299) V* (Outcrossed 2X)

USD971*, *xdh-1(ok3234) IV; sulp-4(rae302) V*

USD997, *xdh-1(ok3234) IV; sulp-4(rae302) V* (Outcrossed 1X)

USD990*, *xdh-1(ok3234) IV; sulp-4(rae319) V*

USD1001, *xdh-1(ok3234) IV; sulp-4(rae319) V* (Outcrossed 1X)

USD1013*, *xdh-1(ok3234) IV; sulp-4(rae320) V*

USD1019*, *xdh-1(ok3234) IV; sulp-4(rae326) V*

*Whole genome sequencing data for these *C. elegans* strains have been deposited at the NIH Sequence Read Archive (SRA) under accession PRJNA1208078.

**CRISPR/Cas9-derived strains.**

USD1042, *sulp-4(rae334) V*

## Chemical mutagenesis and whole genome sequencing

To define *C. elegans* gene activities that were necessary for promoting purine homeostasis, we carried out a chemical mutagenesis screen for mutations that enhanced the penetrance of xanthine stone formation in *xdh-1(ok3234)* mutant *C. elegans* (USD869). *C. elegans* were mutagenized with ethyl methanesulfonate (EMS) using established protocols [22]. Over multiple rounds of mutagenesis, we surveyed ~100,000 mutagenized haploid genomes. To increase the likelihood of identifying a mutation that enhanced xanthine stone formation, we cloned ~600 F2 generation animals that displayed a xanthine stone onto individual NGM petri dishes. F3 generation animals from these ~600 isolates were then screened qualitatively for a population-level increase in xanthine stone penetrance. We demanded that new mutant strains of interest were viable and fertile.

Here, we report the analysis of 5 new mutant strains (USD962, USD971, USD990, USD1013, and USD1019). Each of these strains carried new EMS-induced lesions (*rae299, rae302, rae319, rae320,* or *rae326)* that enhanced the formation of xanthine stones in an *xdh-1* mutant background. Each mutation was recessive; when heterozygous, each lesion caused 5% (*rae302, n* = 42 individuals), 0% (*rae319, n* = 38 individuals), 2% (*rae299, n* = 41 individuals), or 5% (*rae326, n* = 22 individuals) xanthine stone formation in an *xdh-1* mutant background, dramatically reduced when compared to their homozygous counterparts (Fig 2B). *rae320* was never characterized as dominant or recessive.

To further genetically analyze these lesions, we performed complementation analyses of these new mutations. The *rae319* lesion failed to complement *rae302* (100% xanthine stone penetrance, *n* = 71 individuals), *rae299* (87% xanthine stone penetrance, *n* = 38 individuals), and *rae326* (100% xanthine stone penetrance, *n* = 14 individuals). All complementation experiments were performed in an *xdh-1(ok3234)* homozygous mutant genetic background. These results suggest *rae302, rae319, rae299,* and *rae326* all impact the same gene. Complementation studies were not performed on *rae320.*

To identify EMS-induced mutations in our strains of interest, we followed established protocols [59]. Briefly, whole genomic DNA was prepared from *C. elegans* using the Gentra Puregene Tissue Kit (Qiagen) and genomic DNA libraries were prepared using the NEBNext genomic DNA library construction kit (New England Biolabs, MA, USA). DNA libraries were sequenced on an Illumina NovaSeq and deep sequencing reads were analyzed using standard methods on Galaxy, a web-based platform for computational analyses [60]. Briefly, sequencing reads were trimmed and aligned to the WBcel235 *C. elegans* reference genome [61,62]. Variations from the reference genome and the putative impact of those variations were annotated and extracted for analysis [63–65]. All four strains that formed a complementation group possessed novel mutations in the gene *sulp-4,* strongly suggesting that these lesions in *sulp-4* caused the enhanced xanthine stone formation in the *xdh-1* mutant background (Fig 2A and S1 Table). Although the *rae320* lesion found in USD1013 was not analyzed via complementation, whole genome sequence analyses identified a homozygous mutation in *sulp-4*. Thus, we assume that the lesion in *sulp-4* found in USD1013 is also causative for the enhanced xanthine stone formation in the *xdh-1* mutant background (S1 Table). In fact, *rae319* and *rae320* are identical genetic lesions. We know that strains carrying these genetic lesions are not siblings as they were derived from independent rounds of mutagenesis. Whole genome sequencing data for these *C. elegans* strains have been deposited at the NIH SRA under accession PRJNA1208078.

## CRISPR/Cas9 genome editing

Genome engineering using CRISPR/Cas9 technology was performed using established techniques [23,24]. Briefly, 2 guide RNAs were designed and synthesized (IDT, crRNA) that targeted the *sulp-4* locus (5′-agagttagctttgtacaacg-3′ and 5′-atagcacatgatacttccgt-3′). Cas9 (IDT) guide RNA ribonucleoprotein complexes were directly injected into the *C. elegans* germline [23]. Newly induced deletions were identified in the offspring of injected animals using a PCR-based screening approach. The DNA primers used to screen for new deletions were: 5′-gcagagaaactcagagcaacaa-3′ and 5′-gcttggtttggaaactttgg-3′. We were able to isolate and homozygoze *sulp-4(rae334),* a new deletion of *sulp-4* (Fig 2A).

### *C. elegans* transgenesis

Transgenic *C. elegans* carrying extrachromosomal arrays were generated by micro-injecting the gonad of young adult *xdh-1(ok3234); sulp-4(rae319)* double mutant *C. elegans* with an injection mix consisting of the *Psulp-4::SULP-4::GFP* plasmid (20 ng/µl), the *Pmyo-2::mCherry* co-injection marker (2 ng/µl), and the KB+ ladder (78 ng/µl, New England Biolabs) [66]. Three independently derived transgenic strains carrying the extrachromosomal arrays *raeEx118, raeEx119,* or *raeEx120* were isolated and maintained by propagating individual animals based on expression of the fluorescent mCherry protein in the pharynx.

### Determination of xanthine stone penetrance

To determine the percentage of animals that developed a xanthine stone, we cultured WT, mutant, and transgenic *C. elegans* beginning at the L1 stage of development under various growth conditions. Animals were assessed for the formation of a xanthine stone beginning at the L4 stage. Animals were assessed daily through the first 4 days of adulthood except for experiments that used the *cdo-1(mg622)* allele or supplemental cysteine where assays were terminated at day 3 or 2 of adulthood, respectively. *cdo-1(mg622)* and supplemental cysteine caused early lethality that limited the number of individuals that survived per biological replicate. Thus, the assays were shortened to increase the sample size in Figs 1B, 5A, 5B, 6C, and S4. Importantly, all datapoints in a given figure panel were subjected to the same assay conditions and are thus directly comparable. Xanthine stones were determined based upon the presence of exceptionally bright autofluorescent puncta that were opaque when observed by brightfield microscopy. If an individual displayed a stone, it was scored as such and removed from the assay. If an animal did not display a stone, it was counted and moved to a fresh petri dish to prevent contamination from the subsequent generation and allow for assessment on the following day. Thus, if an animal displayed a xanthine stone on any day of the assay, it scored positive and counted towards the penetrance of the phenotype. Xanthine stone penetrance was the percentage of animals that displayed a stone over the course of the assay. If animals went missing or died before the end of the assay, they were not included in the final analyses.

### *C. elegans* larval development and embryonic viability assays

To assay developmental rates, *C. elegans* were synchronized at the first stage of larval development. To synchronize animals, embryos were harvested from gravid adult animals via treatment with a bleach and sodium hydroxide solution. Embryos were then incubated overnight in M9 solution causing them to hatch and arrest development at the L1 stage [67]. Synchronized L1 animals were cultured for 72 hours under standard conditions, and live animals were imaged as described below. Animal length was measured from tip of head to the end of the tail. To define the rate of progression through *C. elegans* development, animals were alternatively scored as being younger than L4 (<L4), L4, or young adults.

To determine the hatching rate of WT and mutant *C. elegans,* we performed synchronized egg lays using young adult animals. Embryos were then scored for hatching ~24 hours after being laid.

### Microscopy

Low magnification bright field and fluorescence images (Figs 1C and S1) were collected using a Nikon SMZ25 microscope equipped with a Hamamatsu Orca flash 4.0 digital camera using NIS-Elements software (Nikon). High magnification differential interference contrast (DIC) and GFP fluorescence images (Figs 2C, 4A, 4B, and S3B–S3D) were collected using a Nikon NiE microscope equipped with a Hamamatsu Orca flash 4.0 digital camera using NIS-Elements software (Nikon). Xanthine stones were visualized and imaged using the EGFP BP (FITC/Cy2) HC Filter Set (Nikon). All images were processed and analyzed using ImageJ software (NIH). All imaging was performed on live animals paralyzed using sodium azide.

## Quantitative PCR (qPCR)

RNA was extracted from synchronized WT, *sulp-4(rae319),* and *osm-8(n1518)* young adult animals using Trizol Reagent per manufacturer's instructions (Invitrogen). Prior to RNA extraction, live *C. elegans* samples were washed and subsequently incubated for one hour in buffer M9 to allow for removal of bacterial contamination. cDNA was then synthesized using the GoScript Reverse Transcriptase System following manufacturer's instructions (Promega). qPCR was performed using a CFX96 Real-Time System (Bio-Rad) and SYBR Green Master Mix following manufacturer's instructions (Applied Biosystems). Relative mRNA levels were calculated using the comparative $C_T$ methods [68]. Forward and reverse amplification primers were *act-1,* 5′-ctcttgccccatcaaccatg-3′ and 5′-cttgcttggagatccacatc-3′; *cdo-1,* 5′-ttcgatgagagaaccggaaag-3′ and 5′-gccattcttagatcctctgtagtc-3′; *hmit-1.1,* 5′-ccattgaagaggtagaaatgc-3′ and 5′-tgtacttcattgtgttgtcc-3′; and *gpdh-1,* 5′-tgcagagattccaggaaaccagg-3′ and 5′-ccctttttgtagcttgccacggag-3′.

## Elemental analyses

For elemental analyses (S, K, Na, Mn, Fe, Cu, Zn, and Mo) of WT or *sulp-4(rae319)* mutant *C. elegans,* 29.9–34.7 mg of tissue (~20,000 freeze-dried young adult *C. elegans*) were digested in 200 μl of concentrated $HNO_3$ at 90 °C for 1 hour. After digestion, samples were diluted to a working volume of 1.4 ml and subsequently diluted an additional 5× (Mn, Fe, Cu, Zn, and Mo) or 200× (S, K, and Na) to produce final samples for elemental analysis. All dilutions were performed with 1% $HNO_3$. Inductively coupled plasma mass spectroscopy (ICP-MS) analysis was performed using an Agilent 8900 triple quad equipped with an SPS autosampler. The system was operated at a radio frequency power of 1,550 W, an argon plasma gas flow rate of 15 L/min, and an Ar carrier gas flow rate of 0.9 L/min. Data were quantified using weighed, serial dilutions of a multi-element standard (CEM 2, (VHG Labs, VHG-SM70B-100) K, Na, Mn, Fe, Cu, and Zn) and a single-element standards for S (Spex CertiPrep, PLS9-2M) and Mo (Spex CertiPrep, CLMO9-2Y).

## NaCl tolerance assay

To assay NaCl tolerance, WT, *xdh-1, sulp-4,* or *osm-8* mutant *C. elegans* were synchronized at the first stage of larval development. To synchronize animals, embryos were harvested from gravid adult animals via treatment with a bleach and sodium hydroxide solution. Embryos were then incubated overnight in M9 solution causing them to hatch and arrest development at the L1 stage [67]. Synchronized L1 animals were cultured under standard conditions for 72 hours or until reaching the young adult stage. Young adult animals were then transferred to NGM plates with various NaCl concentrations (50, 200, 300, 400, 500, or 600 mM NaCl). Animals were scored as alive or dead based on touch responsiveness after a 24-hour exposure to the various NaCl conditions. If animals went missing before the end of the assay, they were not included in the final analyses.

## Supporting information

**S1 Fig. Loss of *sulp-4* enhanced the expressivity of the xanthine stone phenotype displayed by *xdh-1* mutant animals.** Brightfield (left) and fluorescent (right) images of **(A)** *xdh-1(ok3234),* **(B)** *xdh-1(ok3234); sulp-4(rae319),* or **(C)** *osm-8(n1518); xdh-1(ok3234)* mutant adult *C. elegans* cultured on wild-type *E. coli.* Scale bar is 250 μm. Note, *xdh-1(ok3234)* animals displayed are older than their *xdh-1(ok3234); sulp-4(rae319)* or *osm-8(n1518); xdh-1(ok3234)* counterparts which were imaged as day 2 adults. This was necessary to allow us to identify sufficient *xdh-1* single mutant animals displaying a xanthine stone.
(EPS)

**S2 Fig. Inactivating mutations in *sulp* genes other than *sulp-4* did not enhance xanthine stone formation during dietary Moco deficiency. (A)** Wild-type (WT) and *sulp-4(rae319) C. elegans* were cultured on WT (black, Moco+) or

ΔmoaA mutant (red, Moco−) *E. coli* and assessed for the formation of xanthine stones. Note, data points for the WT animals cultured on Moco+ and Moco− *E. coli* are derived from the same experiment that is displayed in Fig 1 B. However, in this analysis the animals were scored until day 4 of adulthood. **(B)** Wild-type and viable *sulp* mutant *C. elegans* were cultured on ΔmoaA mutant (red, Moco−) *E. coli* and assessed for the formation of xanthine stones. Data points represent biological replicates. ****$p < 0.0001$, ordinary one-way ANOVA. If not indicated, data were not statistically different when compared to the wild type. Complete information regarding individuals scored per biological replicate is found in S2 Table. (EPS)

**S3 Fig. Loss of *exc-5* modestly enhanced the formation of xanthine stones in an *xdh-1* mutant background. (A)** *xdh-1(ok3234), exc-5(rh232), exc-5(rh232) xdh-1(ok3234), aex-5(sa23),* and *aex-5(sa23); xdh-1(ok3234)* mutant *C. elegans* were assessed for xanthine stone formation when cultured on wild-type (WT) *E. coli*. Individual data points represent biological replicates. Mean and standard deviation are displayed. ****$p < 0.0001$, ns, $p > 0.05$, ordinary one-way ANOVA. Complete information regarding sample size and individuals scored per biological replicate is found in S2 Table. **(B–D)** Differential interference contrast (top) and fluorescence imaging (bottom) are displayed for (B) WT, (C) *exc-5(rh232)*, or (D) *sulp-4(rae319) C. elegans* expressing the *qpIs11* (Pvha-1::GFP) transgene, which marks the excretory cell. For images of whole animals (left panels), the blue (anterior) and red (posterior) boxes identify the regions displayed in the panels on the right. For images of whole animals (left panels), scale bar is 100 µm. For images of anterior (blue box) and posterior (red box) excretory cell tubule extension, scale bar is 25 µm. (EPS)

**S4 Fig. *cdo-1* was necessary for the enhanced formation of xanthine stones caused by *sulp-4* inactivation during dietary Moco deficiency.** *sulp-4(rae319), sulp-4(rae319); cdo-1(mg622),* and *sulp-4(rae319); moc-1(ok366) cdo-1(mg622)* mutant *C. elegans* were cultured on wild-type (black, Moco+) or ΔmoaA mutant (red, Moco−) *E. coli* and assessed for the formation of xanthine stones over the first 3 days of adulthood. Mean and standard deviation are displayed. Complete information regarding sample size and individuals scored per biological replicate is found in S2 Table. (EPS)

**S5 Fig. Elemental analyses of wild-type and *sulp-4* mutant *C. elegans*.** ICP-MS analysis was performed on extracts from large cultures of young adult wild-type and *sulp-4(rae319)* mutant *C. elegans* to determine the concentrations of **(A)** potassium, **(B)** sulfur, **(C)** iron, **(D)** zinc, **(E)** manganese, **(F)** copper, and **(G)** molybdenum. Individual data points represent biological replicates. Mean and standard deviation are displayed. ns, $p > 0.05$, *t* test. (EPS)

**S6 Fig. *hmit-1.1* and *gpdh-1* were not necessary for the enhanced formation of xanthine stones caused by *sulp-4* inactivation during dietary Moco deficiency.** Relative mRNA expression of **(A)** *hmit-1.1* and **(B)** *gpdh-1* are displayed for total RNA isolated from wild-type, *sulp-4(rae319)*, and *osm-8(n1518)* young adult *C. elegans*. Relative mRNA abundance was determined via the delta-delta $C_T$ method. All transcripts are normalized to *act-1.* Relative mRNA abundance for each transcript was set to one in the wild type. *$p < 0.05$, ns, $p > 0.05$, ordinary one-way ANOVA. **(C)** Wild type and mutant *C. elegans* were cultured on ΔmoaA mutant (red, Moco−) *E. coli* and assessed for the formation of xanthine stones. Individual data points represent biological replicates. Mean and standard deviation are displayed. ns, $p > 0.05$, ordinary one-way ANOVA. Complete information regarding sample size and individuals scored per biological replicate is found in S2 Table. (EPS)

**S1 Table. EMS-induced lesions in *sulp-4* that promoted the formation of xanthine stones in *xdh-1(ok3234)*-mutant *C. elegans*.** (XLSX)

**S2 Table. Raw data and information regarding sample sizes and biological replicates.**
(XLSX)

## Acknowledgments

Some *C. elegans* strains were provided by the CGC, which is funded by the NIH Office of Research Infrastructure Programs (P40 OD010440). We thank the lab of Emily Troemel for providing a *C. elegans* strain carrying the *pnp-1(jy121)* mutation. We thank the lab of Keith Nehrke for providing the *Psulp-4::SULP-4::GFP* plasmid (pTS1). ICP-MS measurements were performed in the OHSU Elemental Analysis Core with partial support from NIH (S10OD028492).

## Author contributions

**Conceptualization:** Jennifer Snoozy, Martina Ralle, Kurt Warnhoff.

**Funding acquisition:** Martina Ralle, Kurt Warnhoff.

**Methodology:** Jennifer Snoozy, Sushila Bhattacharya, Brandon Johnson, Robin R. Fettig, Ashley Van Asma, Chloe Brede, Sophia G. Miller, Martina Ralle.

**Supervision:** Kurt Warnhoff.

**Writing – original draft:** Kurt Warnhoff.

**Writing – review & editing:** Jennifer Snoozy, Martina Ralle, Kurt Warnhoff.

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
