## [Editor Report · Decision Letter 0]

5 Feb 2025

Dear Dr Warnhoff, 

Thank you for submitting your manuscript entitled "XDH-1 inactivation causes xanthine stone formation in C. elegans which is inhibited by SULP-4-mediated anion exchange in the excretory cell" for consideration as a Research Article by PLOS Biology.

Your manuscript has now been evaluated by the PLOS Biology editorial staff as well as by an academic editor with relevant expertise and I am writing to let you know that we would like to send your submission out for external peer review.

Once your full submission is complete, your paper will undergo a series of checks in preparation for peer review. After your manuscript has passed the checks it will be sent out for review. To provide the metadata for your submission, please Login to Editorial Manager (https://www.editorialmanager.com/pbiology) within two working days, i.e. by Feb 07 2025 11:59PM.

Kind regards,

Luke

Lucas Smith, Ph.D.

Senior Editor

PLOS Biology

lsmith@plos.org

---

## [Decision Letter · Decision Letter 1]

24 Mar 2025

Dear Dr Warnhoff,

Thank you for your patience while your manuscript "XDH-1 inactivation causes xanthine stone formation in C. elegans which is inhibited by SULP-4-mediated anion exchange in the excretory cell" was peer-reviewed at PLOS Biology. It has now been evaluated by the PLOS Biology editors, an Academic Editor with relevant expertise, and by several independent reviewers. 

In light of the reviews, which you will find at the end of this email, we would like to invite you to revise the work to thoroughly address the reviewers' reports.

As you will see below, the reviewers appreciate that the study is thorough and the findings are novel, but they raise a number of issues with the strength or relevance of the conclusions and with the presentation of the findings. We think the reviewer comments will need to be thoroughly addressed before we can consider the manuscript further for publication. 

Given the extent of revision needed, we cannot make a decision about publication until we have seen the revised manuscript and your response to the reviewers' comments. Your revised manuscript is likely to be sent for further evaluation by all or a subset of the reviewers.

**IMPORTANT - SUBMITTING YOUR REVISION**

*Re-submission Checklist*

*Published Peer Review*

*PLOS Data Policy*

*Blot and Gel Data Policy*

Sincerely,

Luke

Lucas Smith, Ph.D.

Senior Editor

PLOS Biology

lsmith@plos.org

REVIEWS:

Reviewer #1: The authors utilize C. elegans xdh-1 mutants as a model for Xanthinuria, an inborn error of purine metabolism. In the absence of molybdenum cofactor or in xdh-1 mutants, C. elegans forms xanthine stones in the lumen of the posterior intestine. The number of xdh-1 mutants that form stones is low so the authors performed a forward genetic screen for mutants that enhance stone formation in the xdh-1 background. Multiple alleles of the SLC26 transporter sulp-4 were identified. Sulp-4 appears to only be expressed in the excretory cell, suggesting it regulates stone formation in the intestine at a distance. Further genetic analysis shows that inhibition of enzymes involved in sulfur amino acid catabolism (cdo-1, cth-2) suppress stone formation caused by sulp-4;xdh-1. Because cdo-1 and cth-2 are involved in synthesis of the osmolyte taurine, the authors propose that sulp-4 animals are in a state of osmotic stress (due to an inability to secrete sulfure) and that this drives flux through the cdo-1/cth-2 pathway to produce taurine as an adaptive response. Data from a bonafide osmotic stress mutant and qPCR of validated osmotic response genes suggest some level of osmotic stress may be occurring in sulp-4 mutants.

I find this work to be thorough and genetically rigorous. However, some of the data interpretation was confusing and seems to ignore simple hypotheses for more complex ideas. Also, the paper lacks an integrated model that accounts for all of the data presented so I found it very hard to interpret the overall significance. Some of my questions are noted below:

1. A model that integrates the data would be very helpful to readers and should be included.

2. It seems to me that a simpler model than the 'osmotic stress' model proposed by the authors is that whole animal sulfur levels, derived in part from cdo-1/cth-2 catabolic pathways, increase when sulfur excretion is blocked in sulp-4 mutants. Excess sulfur drives stone formation. Loss of cdo-1/cth-2 lowers sulfur generation and blocks stone formation. Was this model tested and disproven in some way?

3. What do the authors mean by the sulp-4 mutant causing osmotic stress? Do they mean that blocking sulfate transport in the excretory cell increase overall solute levels enough to osmotically stress the animal, which activates the osmotic stress response, like exposing animals to hypertonic stress or knocking out osm-8? If so, osm-8 type mutants have a very striking osmotic stress resistance (Osr) phenotype. Do sulp-4 mutants have an Osr phenotype like osm-8? 

4. A further prediction of this osmotic stress model is that growing animals in a hypertonic environment should also drive stone formation. Was that tested? If so, is it cdo-1/cth-2 dependent?

5. Can xanthine stone formation occur in vitro? If so, that could be an interesting setting in which to directly test the sulfur hypothesis.

6. Is there any evidence that C. elegans accumulates taurine as an osmolyte in response to osmotic stress? Also, where taurine is used as an osmolyte in mammalian cells, it is usually accumulated via a membrane transporter, not so much by biosynthesis. 

7. Osm8 mutants contain high levels of glycerol, which creates an osmotic driving force for water reabsorption. Could the effect of osm-8 simply be due to dehydration of the intestinal lumen, which would increase hypoxanthine concentration and promote stone formation? 

8. How could hypoxanthine accumulate to high enough levels in the intestinal lumen to form stones, given that the worm defecates and flushes the intestinal lumen every 45 seconds? Are xdh-1,, sulp-4, or other stone-inducing mutants defecation-defective?

9. The pictures of the excretory cell in figure S3 are not of sufficient quality to determine whether or not sulp-4 alters cell morphology. The cysts are notable in exc-5 and are clearly not present in sulp-4. However, I can not determine if the excretory cell tubules are fully extended in either WT or sulp-4.

10. The qPCR data are somewhat concerning and do not support the conclusion on line 404-405 that sulp-4 causes increased expression of gpdh-1 and hmit-1.1. The authors cannot claim this as evidence while simultaneously claiming the changes are not statistically significant. Unfortunately, I think the 1 hour starvation prior to RNA isolation is to blame here, given that gpdh-1 and hmit-1.1 are well established by multiple labs to be strongly upregulated by osm-8 and other Osr mutants, as well as hypertonic stress, a finding that was not reproduced using the authors starvation protocol. If the authors wish to make this claim, the experiment should be repeated, probably without the 1 hour starvation.

Reviewer #2: In this manuscript, Snoozy et al demonstrate that specific C elegans mutants can develop xanthine stones, similar to those observed in humans with xanthine dehydrogenase (XDH-1) deficiency. While the loss of xdh-1, the C. elegans homolog of XDH-1, alone resulted in low penetrance of stone formation, the authors determined that loss of sulp-4, a putative anion transporter, enhanced xdh-1 stone formation significantly. Moreover, the author's data indicate sulp-4 may be a negative regulator of osmotic stress, and perturbations in this pathway contribute to stone moderation. Generally, the paper is well-written, and the experimental design is sound. 

The work is novel; however, it seems incomplete as there is no description of the impact of stone formation on any aspect of worm biology. While it is interesting that the authors have documented stone formation in C. elegans for the first time, it is unclear how pertinent this system is. For example, the authors state that the role of SLC26 transporters in stone formation has already been documented. Also, are a large percentage of xanthinuria cases in humans due to unknown genetic mutations that could be illuminated by genetic screens in C. elegans? As the formation of stones is linked to dietary intake, this seems unlikely. Perhaps, this interpretation is completely inaccurate and if so, it would then be critical for the authors to stress the impact of this model beyond the discovery of xanthine stones in C. elegans. 

Major Revisions:

1) Figure 1A should be replaced with a schematic of Moco synthesis, moc genes, and interactions with cth-2 and cdo-1. Please describe cth-2 and cdo-1 and their interaction with Moco when these genes are first introduced, not in the discussion. Fig 1A fits better with the data presented in Fig 3 and should be moved there. 

2) Quantifying worm length indicates that the mutants presented in Fig 2D, Fig 3D, and Fig 5C are shorter than N2 72 hours post-L1 plating, but it does not directly address their larval stage. If the authors want to state that larval development is affected, they must quantify the number of animals that reach the L4 stage 48 hours post-L1 plating. 

3) Only one biological replicate for Sup Fig 2B is shown. Published results from experiments should be performed in triplicate at a minimum. 

4) Along with including more biological replicates for Sup Fig2B, demonstrating that overexpression of another SLC26 homolog does not rescue the stone formation in xdh-1;sulp-4 will strengthen the specific function of sulp-4 in contrast to general disruption of anion transport due to overexpression of sulp-4. 

5) The number of haploid genomes screened in the xdh-1 enhancer screen and the total number of mutations and complementation groups should be provided. 

6) Provide a model figure tying the expression and function of sulf-4, osm-8 and xdh-1 to stone formation in the intestine.

Other issues/questions:

1) What are the three other Moco-requiring enzymes and their function? 

2) Has xdh-1 been shown to require Moco for its enzymatic function, or is this just assumed based on sequence homology to XDH? Were is xdh-1 expressed?

3) re319 and re320 have the same mutation and display different penetrance of stones. Is the annotation correct in Fig 2A?

4) To state that xdh-1 and sulp-4 act in parallel, one needs to determine if there is a significantly higher penetrance in stones in an xdh-1;sulp-4 double compared to the singles. Figure 2B does not do this, as the single mutants are not shown. This statement is confirmed by piecing together data from multiple figures (Fig 1B, Sup Fig 2A, and Fig 2B), but it needs to be presented more clearly. 

5) Where is pnp-1 expressed, and how does this relate to xdh-1, xanthine, and stone formation? 

6) Can exogenous supplied taurine reverse the suppression of stone formation observed in xdh-1;sulp-4;cdo-1 mutant? This would strengthen the connection to osmotic stress. 

7) Where is osm-8 expressed? A quick search indicates that it is expressed in the hypodermis. How does this relate to stone formation in the intestine and the function of sulp-4 in the excretory cell?

Reviewer #3: PBIOLOGY-D-25-00292R1

XDH-1 inactivation causes xanthine stone formation in C. elegans which is inhibited by SULP-4-mediated anion exchange in the excretory cell

Jennifer Snoozy, Sushila Bhattacharya, Robin R. Fettig, Ashley Van Asma, Chloe Brede, Kurt Warnhoff

General.

This manuscript examining the interaction of XDH-1 and SULP-4 in C. elegans nicely demonstrated several phenotypes highlighting metabolic dependence. Importantly, Psulp-4::SULP-4::GFP worms do not develop xanthine crystals demonstrating that labeled SULP-4 can replace function in sulp-4 mutants. The authors have provided a fairly thorough analysis of possible pathways that could account for this interaction including sulfur containing amino acid metabolism and osmolarity changes. 

There are a few points that the authors should consider.

Major.

1) Method explanations 

a) Stone counts / %: Methods should be more clear as to what a positive stone count is and how "% penetrance" is calculated. It seems that even one crystal counts that worm as positive. The % penetrance then comes from the number withing that group that have "a crystal"

b) C. elegans larval delay: it is implied but not stated (especially for the non- C. elegans reader) that larval length is used as a surrogate for larval delay. Please reference or explain more overtly

2) Discussion. 

a) Why is xanthine accumulating in xdh-1; sulp-4 animals if SULP-4 is a chloride and sulfate transporter? What is the connection? For example, xanthine is typically degraded to uric acid (urate) which can also form stones. Is there evidence of uric acid crystals forming? Or is metabolism further in C. elegans? (e.g., allantoin, urea, NH3 ?). Considering this further metabolism may implicate why defective sulp-4 results in a back up of xanthine. Why would removing sulfate from the metabolic steps prevent an accumulation of xanthine?

b) Is there an effect of worm size per se on xanthine crystal formation? Smaller worms would have a tendency to for crystals more readily due to smaller spaces for xanthine to accumulate, Figures 2 and 5 would seem consistent with this interpretation. Are there reasons why these would be dissociated?

3) Supplemental figures. These 4 figures would seem appropriate in the main text as each has important data that are presented in Results.

Minor.

* Suggest using "xanthine crystal" rather than "xanthine stone"

* Microscope / filter setting for xanthine crystal autofluorescence? Are these FITC/GFP settings or other?

* Check background on stone types. There are more recent assessments than 1954

* Is size of xanthine crystal significant? For example, Supp Fig 1 shows that osm-8;xdh-1 crystals are much smaller than those observed with xdh-1;sulp-4.

* In testing for xanthine crystal penetrance (Supp Fig 2), why are sulp-3 and sulp-6 missing? If viable strains are not available, this should be stated around line 196-197.

* Perhaps a Table of C elegans genes and functions would help a more general reader

* No institutional association for Ashley Van Asma

---

## [Decision Letter · Decision Letter 2]

27 Aug 2025

Dear Dr Warnhoff,

Thank you for your patience while we considered your revised manuscript "XDH-1 inactivation causes xanthine stone formation in C. elegans which is inhibited by SULP-4-mediated anion exchange in the excretory cell" for publication as a Research Article at PLOS Biology. This revised version of your manuscript has been evaluated by the PLOS Biology editors, the Academic Editor and two of the original reviewers.

Based on the reviews, we are likely to accept this manuscript for publication, provided you satisfactorily address the remaining points raised by the reviewers. Additionally, please also make sure to address the following data and other policy-related requests.

**IMPORTANT: Along with the reviewer comments, please address the following points.

1) DATA: Thank you for providing your genome sequencing data on SRA. It appears these data are set to private, as I was not able to access them w/ the provided accession number. Please note that these data will need to be public at the time of publication. Also, in the meantime, can you provide me with a reviewer token so I can ensure these data meet our requirements? (sorry if I missed this somewhere!) 

2) CODE: Per journal policy, if you have generated any custom code during the course of this investigation, please make it available without restrictions. Please ensure that the code is sufficiently well documented and reusable, and that your Data Statement in the Editorial Manager submission system accurately describes where your code can be found. 

We expect to receive your revised manuscript within two weeks. 

*Published Peer Review History*

*Press*

Sincerely,

Luke

Lucas Smith, Ph.D.

Senior Editor

lsmith@plos.org

PLOS Biology

Reviewer remarks:

Reviewer #1: The authors have performed extensive revisions of the study to address the many comments and suggestions of the reviewers. These revisions greatly enhance the conclusions and present the data in a way that is much more comprehensible. I particularly appreciate the models in Figure 1 and 7. This nicely sets up the metabolic pathway and puts the results into a clear and functional context. I also appreciate the new data examining whole animal sulfur. However, I do have one remaining concern related to the last bit of data on gpdh-1 and hmit-1.1. The authors use gpdh-1 and hmit-1.1 deletion alleles to test the role of glycerol and myo-inositol in stone accumulation. Loss of gpdh-1 alone does not reduce steady state glycerol levels (Lamitina et al 2006). The authors would need to delete both gpdh-1 and gpdh-2 to reduce glycerol levels (and even that only reduces high salt induced glycerol accumulation by ~60%). I am not aware of any data examining myo-inositol levels in C. elegans, much less if loss of hmit-1.1 function reduces myo-inositol levels. This is also complicated by the presence of 3 hmit genes in worms, which could also be involved in regulating myo-inositol levels. So statements about the specific roles of glycerol, myo-inositol, or other osmolytes (or lack thereof) are not warranted. The authors should change their conclusion to address these limitations.

Reviewer #2: The authors have done great work to address the reviewers' comments by refining the text and including additional experiments that together improve the clarity and rigor of the work. 

My only comments regard the author's statement that for Fig 2D, 3B, and 5C, the previous data using worm length as a proxy for larval development has been replaced with quantification of time to reach the young adult stage.

1) The data in these figures has not been replaced

2) The figure legends have not been updated to reflect this change

3) The methods section still includes a description of how the worm length has been measured. 

This should be corrected before publication.

---

## [Editor Report · Decision Letter 3]

11 Sep 2025

Dear Dr Warnhoff,

Thank you for the submission of your revised Research Article "XDH-1 inactivation causes xanthine stone formation in C. elegans which is inhibited by SULP-4-mediated anion exchange in the excretory cell" for publication in PLOS Biology and thank you for addressing the last reviewer and editorial points in this revision. On behalf of my colleagues and the Academic Editor, Mark J Alkema, I am pleased to say that we can in principle accept your manuscript for publication, provided you address any remaining formatting and reporting issues. These will be detailed in an email you should receive within 2-3 business days from our colleagues in the journal operations team; no action is required from you until then. Please note that we will not be able to formally accept your manuscript and schedule it for publication until you have completed any requested changes.

PRESS

We frequently collaborate with press offices. If your institution or institutions have a press office, please notify them about your upcoming paper at this point, to enable them to help maximize its impact. If the press office is planning to promote your findings, we would be grateful if they could coordinate with biologypress@plos.org. If you have previously opted in to the early version process, we ask that you notify us immediately of any press plans so that we may opt out on your behalf.

Sincerely, 

Luke

Lucas Smith, Ph.D.

Senior Editor

PLOS Biology

lsmith@plos.org